# Mixture-of-World Models: Scaling Multi-Task Reinforcement Learning with Modular Latent Dynamics

**Boxuan Zhang**[†] **Weipu Zhang**[◇] **Zhaohan Feng**[†] **Wei Xiao**[†] **Jian Sun**[†] **Jie Chen**[⋆] **Gang Wang**[†*]

[†]National Key Lab of Autonomous Intelligent Unmanned Systems, Beijing Institute of Technology
[⋆]Department of Control Science and Engineering, Harbin Institute of Technology
[◇]Jiangxing Intelligence Inc.
{zhangboxuan,zhfeng,xiaowei,sunjian,chenjie,gangwang}@bit.edu.cn
weipuzhang.academic@gmail.com

## Abstract

A fundamental challenge in multi-task reinforcement learning (MTRL) is achieving sample efficiency in visual domains where tasks exhibit significant heterogeneity in both observations and dynamics. Model-based RL (MBRL) offers a promising path to sample efficiency through world models, but standard monolithic architectures struggle to capture diverse task dynamics, leading to poor reconstruction and prediction accuracy. We introduce the mixture-of-world models (MoW), a scalable architecture that integrates three key components: i) modular VAEs for task-adaptive visual compression, ii) a hybrid Transformer-based dynamics model combining task-conditioned experts with a shared backbone, and, iii) a gradient-based task clustering strategy for efficient parameter allocation. On the Atari 100K benchmark, **a single MoW agent** (trained once over Atari 26 games) achieves a mean human-normalized score of $110.4\%$, competitive with the score $114.2\%$ achieved by the recent STORM—an ensemble of 26 task-specific models—while requiring $50\%$ fewer parameters. On Meta-World, MoW attains a $74.5\%$ average success rate within 300K steps, establishing a new state-of-the-art. These results demonstrate that MoW provides a scalable and parameter-efficient foundation for generalist world models.

## 1 Introduction

The pursuit of sample-efficient reinforcement learning (RL) has led to significant progress in complex domains, from game-playing (Silver et al., 2016; Hansen et al., 2024) to robotic control (Ye et al., 2020). However, these advances are often confined to single-task settings, where the learned policies lack generalization (Yu et al., 2020a; Feng et al., 2026). Multi-task RL (MTRL) aims to overcome this by learning a unified policy across a diverse task suite (Yang et al., 2020). While model-free methods have been extended to MTRL (Xing et al., 2025), they inherit the poor sample efficiency of their single-task counterparts (Mnih et al., 2015; Schulman et al., 2017), limiting their applicability in real-world scenarios where data is scarce (Xia et al., 2018; Huang et al., 2025).

In single-task settings, model-based RL (MBRL) has emerged as a path to achieving greater sample efficiency. World models learn compact latent representations of environment dynamics, enabling policy optimization entirely through "latent imagination" (Ha & Schmidhuber, 2018; Hafner et al., 2019; 2025). These methods typically combine a variational autoencoder (VAE) with a latent recurrent state-space model (RSSM) using e.g., a recurrent neural network (RNN) (Hafner et al., 2019) or Transformer (Vaswani, 2017), and have shown remarkable success (Zhang et al., 2025; Hafner et al., 2025; Micheli et al., 2022; Zhang et al., 2023).

Scaling world models to multi-task, visual domains presents a fundamental challenge: the inherent visual and dynamic heterogeneity across tasks. A single world model must simultaneously capture

---

[*]Corresponding author

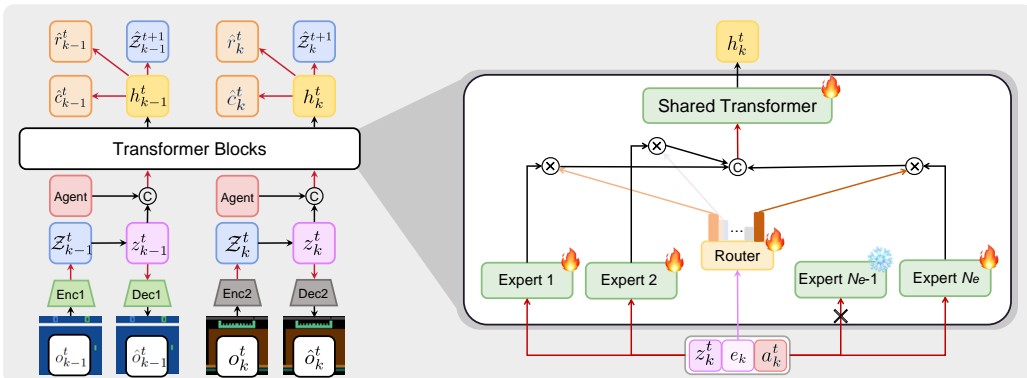

Figure 1: Overview of the mixture-of-world models (MoW) architecture. Task-specific observations are encoded through specialized VAEs, with dynamics modeled by a mixture-of-Transformer experts routed via task embeddings. The design enables modular latent dynamics handling while maintaining parameter efficiency. Here, at time $t$ for task $k$, $o_k^t$, $r_k^t$, $c_k^t$ and $a_k^t$ denote the high-dimensional observation, reward, termination flag, and action, respectively. The stochastic representation $z_k^t$ is sampled from the distribution $\mathcal{Z}_k^t$, which is encoded from the observation $o_k^t$. The hidden state $h_k^t$ is learned by the hybrid Transformer architecture, while $e_k$ represents the learnable task embedding and $N_e$ is the number of experts.

disparate visual features while accurately predicting task-specific dynamics—objectives that often conflict in practice, straining the capacity of monolithic architectures (Yu et al., 2020b). Simply conditioning a shared model on task identity often proves insufficient without abundant expert data (Hansen et al., 2024). Crucially, for visual MBRL, the model must achieve high-fidelity reconstruction to enable effective policy learning in the latent space—a challenge that remains largely unaddressed in MTRL.

In this work, we propose the mixture-of-world models (MoW), a novel architecture designed for parameter- and sample-efficient MTRL in visual domains. The MoW explicitly addresses multi-task heterogeneity through a modular design. It employs a set of task-specific VAEs for visual encoding and a spatial-temporal dynamics core composed of a mixture of expert Transformers, guided by learned task embeddings. To ensure effective module specialization, we introduce a task prediction loss that enhances task discrimination in the latent space and an expert balance loss to prevent module collapse. Furthermore, we develop a gradient-based task clustering strategy during a warm-up phase to share components across tasks, significantly improving parameter efficiency.

We evaluate MoW on two challenging benchmarks: the Atari 100K benchmark (discrete control) and Meta-World (continuous control). Our results demonstrate that a **single MoW agent** achieves human-normalized score performance competitive with an ensemble of 26 task-specific models on Atari while using half the parameters, and a $74.5\%$ success rate on Meta-World within 300K steps, setting a new state-of-the-art. Ablation studies confirm the contribution of each component.

Our main contributions are summarized below.

- **The MoW architecture**: A mixture-of-world models with modular VAEs and transformer experts that captures diverse task dynamics while maintaining reconstruction fidelity.

- **Specialized losses**: Task prediction and expert balance losses that enable effective task discrimination and balanced utilization across expert components.

- **Gradient-based clustering**: A warm-up strategy that optimizes parameter sharing by clustering tasks based on gradient similarity, enhancing model efficiency.

- **State-of-the-art performance**: Empirical validation on Atari 100K and Meta-World demonstrating superior sample efficiency and parameter scaling.

## 2 RELATED WORK

### 2.1 MULTI-TASK REINFORCEMENT LEARNING

Multi-task reinforcement learning seeks to develop agents capable of generalizing across a diverse set of tasks by leveraging shared experience and representations. A dominant paradigm in online MTRL employs task-conditioned policies, where a unified network or value function is conditioned on task identifiers or embeddings (Xu et al., 2020; Sodhani et al., 2021b). This approach has successfully adapted celebrated model-free algorithms like PPO (Schulman et al., 2017) and SAC (Haarnoja et al., 2018) to multi-task settings, though they inherit the sample inefficiency inherent to model-free RL methods. Offline MTRL methods, such as those based on decision Transformers (Lee et al., 2022; Kong et al., 2025), learn from fixed datasets of expert demonstrations. While effective within their data constraints, these approaches cannot improve through online interaction and are typically limited to state-based inputs, where errors in task conditioning can critically degrade performance when extended to visual inputs.

The mixture-of-experts (MoE) architecture, widely successful in scaling large language models to trillions of parameters by virtue of their modular, sparsely activated design (Fedus et al., 2022; Lepikhin et al., 2020), has recently been applied to MTRL. In such models, only a small subset of experts is invoked for each input token, enabling dramatic increases in model capacity without proportional growth in computational cost. Methods like MOORE (Hendawy et al., 2023) and D2R (He et al., 2024) incorporate MoE into SAC, improving knowledge sharing and specialization. However, these approaches remain evaluated primarily on low-dimensional state spaces and are hampered by the sample inefficiency of model-free learning. The significantly greater challenge of high-dimensional visual inputs-with their complex perception and dynamics-remains largely unaddressed by existing model-free MoE methods.

### 2.2 WORLD MODELS FOR REINFORCEMENT LEARNING

World models have emerged as a powerful approach for improving sample efficiency in RL by enabling learning from imagined trajectories. Early work like SimPLe (Kaiser et al., 2019) demonstrated the potential of learned models for policy learning. The Dreamer series (Hafner et al., 2019; 2025) advanced this paradigm with recurrent state-space models that support long-horizon latent imagination. Recently, Transformers have shown strong performance as world models due to their capacity for long-range dependency modeling. IRIS (Micheli et al., 2022) and $\Delta$-IRIS (Micheli et al., 2024) employ VQ-VAEs (Van Den Oord et al., 2017) for discrete visual tokenization, while STORM (Zhang et al., 2023) achieves state-of-the-art results on Atari with a customized Transformer architecture. DIAMOND (Alonso et al., 2024) explores diffusion-based world models operating directly in pixel space, improving visual fidelity.

Critically, these advances all focus on single-task RL settings. While TD-MPC2 (Hansen et al., 2024) extends model-based principles to multi-task domains, it operates on low-dimensional state inputs rather than more challenging visual observations and relies on expert data. The challenge of building world models that simultaneously handle diverse visual inputs and dynamics across multiple tasks—while maintaining reconstruction fidelity for effective latent imagination—remains open. Our work directly addresses this gap by introducing a modular architecture specifically designed for multi-task visual world models.

## 3 METHOD

We consider a multi-task setting with $K$ tasks collectively denoted by $\mathcal{T} = \{T_1, T_2, \cdots, T_K\}$. Each task $T_k$ is a partially observable Markov decision process (POMDP) defined by the tuple $\mathcal{M}_k = (\mathcal{O}_k, \mathcal{A}_k, \mathcal{P}_k, \mathcal{R}_k, \gamma_k)$, where $\mathcal{O}_k$ is the observation space, $\mathcal{A}_k$ is the action space, $\mathcal{P}_k$ is the transition dynamics, $\mathcal{R}_k$ is the reward function, and $\gamma_k \in [0, 1)$ is the discount factor. The objective is to learn a single policy $\pi$ that maximizes the expected discounted return across all tasks: $\sum_{k=1}^{K} \mathbb{E}_{(o_k, a_k) \sim \pi, \mathcal{P}_k} \left[ \sum_{t=1}^{\infty} \gamma_k^{t-1} \mathcal{R}_k(o_k^t, a_k^t) \right]$.

From the preceding observations and discussions, our key insight is that constructing a mixture of world models across heterogeneous tasks, along with performing imagination and training in paral-

lel, enables the development of a more generalizable multi-task agent while significantly possessing the parameter scalability.

## 3.1 ARCHITECTURE OVERVIEW

The MoW consists of two primary modules: a *perceptual module* that encodes high-dimensional visual observations into a compact latent space, and a *temporal module* that learns task-adaptive dynamics within this latent space. The overall architecture is depicted in Figure 1.

### 3.1.1 PERCEPTUAL MODULE

The perceptual module encodes each image observation $o_k^t$ into a stochastic representation $z_k^t$ using a categorical VAE. Each task $T_k$ is assigned a specific encoder-decoder pair $(q_{\phi,i_k}, p_{\phi,i_k})$, where the index $i_k \in [1, N_m]$ is determined by a gradient-based clustering procedure detailed in Section 3.6. To avoid an excessive number of hyperparameters, $N_m$ is not only the number of VAEs, but also the number of distribution, reward, continuation predictors and critic networks mentioned later (all components with the subscript $i_k$, as the predictors and VAEs are corresponding in task $k$). A learnable task embedding $e_k$ (with $\|e_k\|_2 = 1$) conditions the encoding and decoding processes. The stochastic representation $z_k^t$ is sampled from the categorical posterior distribution $\mathcal{Z}_k^t$ (comprises 32 categories, each with 32 classes) to serve as a compact representation of the original observation with the straight-through gradient trick (Bengio et al., 2013):

$$\text{Posterior:} \quad z_k^t \sim q_{\phi,i_k}(z_k^t \mid o_k^t, e_k) = \mathcal{Z}_k^t$$
$$\text{Reconstruction:} \quad \hat{o}_k^t \sim p_{\phi,i_k}(\hat{o}_k^t \mid z_k^t, e_k). \tag{1}$$

### 3.1.2 TEMPORAL MODULE

The temporal module models the dynamics in the latent space, which takes the sequence of the stochastic representations and actions as input, and predicts the next latent state, reward, task index, and continuation flag. The module is composed of a mixture of expert Transformers and a shared Transformer, while the variables are also conditioned by the task embedding $e_k$.

**Input Representation.** The stochastic representation $z_k^t$ and action $a_k^t$ are first concatenated via a multi-layer perceptron (MLP) $m_{\phi,j}$ to a unified token $m_{j,k}^t$ for each expert $j$, where $j = 1, 2, \cdots, N_e$ index the activated expert Transformers.

$$m_{j,k}^{1:t} = m_{\phi,j}(z_k^{1:t}, a_k^{1:t}, e_k). \tag{2}$$

**Expert Selection.** We employ a task-level router function that takes the task embedding $e_k$ as the input, and produces a set of expert indices $J_k$ with their corresponding weights $W_k$ via a TopK operation applied to a softmax output of an MLP.

$$S_k = \text{Softmax}\left(\text{MLP}(e_k)\right),$$
$$W_k, J_k = \text{TopK}\left(S_k, \text{topk} = n_k\right), \tag{3}$$

where $S_k$ denotes the task-to-expert affinity scores, TopK selects the $n_k$ largest scores, $J_k = [j_k^1, j_k^2, \cdots, j_k^{n_k}]$ denote the activated expert Transformer indices, and $W_k = [0, \cdots, w_k^1, \cdots, w_k^2, \cdots, w_k^{n_k}, \cdots, 0]$ is the normalized weight vector of activated experts, where the position of $w_k$ indicates the activation of the corresponding expert. The selection of $i_k$ is determined during the warmup stage by clustering the gradient vectors and is fixed throughout training. In contrast, the expert indices $J_k$ are dynamically determined by a TopK function conditioned on the current task embedding, and changes over time as the world model is trained in an end-to-end manner but remains fixed in a single training step.

**Expert and Shared Transformers.** The tokens are processed by the activated expert Transformers $f_{\phi,j}$ for $j \in J_k$. The outputs of the experts are concatenated to form a token $l_k^{1:t}$, which is then passed to the shared Transformer $F_\phi$ to produce the hidden state $h_k^t$. The expert Transformers specialize in modeling task-specific dynamics, while the shared Transformer captures common knowledge across the entire domain.

$$l_k^{1:t} = \text{concat}\left[f_{\phi,j_k^1}(m_{j_k^1,k}^{1:t}), \cdots, f_{\phi,j_k^{n_k}}(m_{j_k^{n_k},k}^{1:t})\right],$$
$$h_k^{1:t} = F_\phi(l_k^{1:t}, e_k). \tag{4}$$

**Prediction Heads.** The hidden state $h_k^t$ is used to predict the next latent state distribution, reward, and continuation flag. Additionally, an auxiliary task predictor head is used to encourage task-discriminative feature extraction:

$$
\begin{aligned}
\hat{\mathcal{Z}}_k^{t+1} &= g_{\phi,i_k}^D(\hat{z}_k^{t+1} \mid h_k^t, e_k), \quad \hat{r}_k^t = g_{\phi,i_k}^R(h_k^t, e_k), \\
\hat{c}_k^t &= g_{\phi,i_k}^C(h_k^t, e_k), \qquad\qquad \hat{k} = g_\phi^T(h_k^t).
\end{aligned}
\tag{5}
$$

An underlying intuition is that the expert selection should remain relatively stochastic in the early stages of training to facilitate balanced expert utilization. As training progresses and converges, the selection should gradually become more deterministic; otherwise, frequent switching of experts may destabilize the end-to-end optimization. To embody this intuition, we anneal the temperature coefficient in the softmax function progressively toward 1 during training to avoid excessively low temperatures that result in nearly one-hot distributions, which would hinder experts' collaboration.

## 3.2 DESIGN RATIONALE FOR TASK-LEVEL ROUTING

We adopt task-level routing instead of token-level routing, in other words, we use task embeddings $e_k$ rather than mixture tokens $m_{j,k}^t$ to select experts, based on the following considerations. First, in conventional Transformer architectures, MoE is typically applied to the feed-forward networks rather than the self-attention layers. This design implies that the MoE module primarily models sparse transformations between data features, rather than capturing the temporal dependencies that self-attention layers focus on. To address this, we decouple the MoE mechanism from the Transformer architecture and apply it externally, enabling MoW to learn task-specific dynamics across different tasks through the mixture of transformer backbones.

Second, if expert Transformers are selected based on the mixed token, each token along the training trajectory may activate different experts, even within the same task. Such dynamic routing results in each expert observing only partial sequences, leading to the fragmented learning of task dynamics. As a consequence, the experts may fail to effectively model the full temporal structure of the task. In contrast, routing based on task embeddings ensures consistent expert activation within each task, allowing individual experts to capture more complete and coherent task dynamics.

## 3.3 WORLD MODEL LEARNING

We train the MoW in an end-to-end manner using a self-supervised learning paradigm, leveraging data from all tasks jointly. For task $T_k$, the total loss is given by

$$
\mathcal{L}_k(\phi) = \sum_t \left[ \mathcal{L}_{t,k}^{\text{rec}}(\phi) + \mathcal{L}_{t,k}^{\text{rew}}(\phi) + \mathcal{L}_{t,k}^{\text{con}}(\phi) + \mathcal{L}_{t,k}^{\text{task}}(\phi) + \beta_1 \mathcal{L}_{t,k}^{\text{dyn}}(\phi) + \beta_2 \mathcal{L}_{t,k}^{\text{rep}}(\phi) \right],
\tag{6}
$$

with hyperparameters $\beta_1 = 0.5$ and $\beta_2 = 0.1$ fixed, where $\mathcal{L}_{t,k}^{\text{rec}}(\phi)$ denotes the reconstruction loss of the original image, $\mathcal{L}_{t,k}^{\text{rew}}(\phi)$ the prediction loss of the reward, $\mathcal{L}_{t,k}^{\text{con}}(\phi)$ the prediction loss of the continuation flag, and $\mathcal{L}_{t,k}^{\text{task}}(\phi)$ represents the prediction loss of task $T_k$:

$$
\begin{aligned}
\mathcal{L}_{t,k}^{\text{rec}}(\phi) &= \|\hat{o}_k^t - o_k^t\|_2, & \mathcal{L}_{t,k}^{\text{rew}}(\phi) &= \text{SymlogCrossEnt}(\hat{r}_k^t, r_k^t), \\
\mathcal{L}_{t,k}^{\text{con}}(\phi) &= \text{BinaryCrossEnt}(\hat{c}_k^t, c_k^t), & \mathcal{L}_{t,k}^{\text{task}}(\phi) &= \text{CrossEnt}(\hat{k}, k).
\end{aligned}
\tag{7}
$$

Introducing auxiliary predictors and their associated losses has proven effective for shaping the representation of latent states (Burchi & Timofte, 2025). Therefore, we incorporate an additional task prediction loss $\mathcal{L}_{t,k}^{\text{task}}(\phi)$ to explicitly guide the training of hidden states $h_k^t$ toward being task-discriminative. This ensures that the hidden states contain sufficient information for task identification, thereby enhancing the accuracy of the imagination process.

The losses $\mathcal{L}_{t,k}^{\text{dyn}}(\phi)$ and $\mathcal{L}_{t,k}^{\text{rep}}(\phi)$ train the dynamics predictor network to predict the next stochastic distributions from the hidden states by minimizing the Kullback–Leibler (KL) divergence between the predictor output distribution $\hat{\mathcal{Z}}_k^{t+1} = g_{\phi,i_k}^D(\hat{z}_k^{t+1}|h_k^t)$ and the next encoder representation $\mathcal{Z}_k^t = q_{\phi,i_k}(z_k^t|o_k^t, e_k)$:

$$
\begin{aligned}
\mathcal{L}_{t,k}^{\text{dyn}}(\phi) &= \max\left(1, \text{KL}\left[\text{sg}(\mathcal{Z}_k^{t+1}) \| \hat{\mathcal{Z}}_k^{t+1}\right]\right), \\
\mathcal{L}_{t,k}^{\text{rep}}(\phi) &= \max\left(1, \text{KL}\left[\mathcal{Z}_k^{t+1} \| \text{sg}(\hat{\mathcal{Z}}_k^{t+1})\right]\right),
\end{aligned}
\tag{8}
$$

Imagined by multi-task STORM          Imagined by MoW

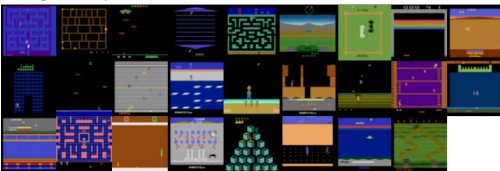

Figure 2: The MoW enables precise modeling and reconstruction of task-specific dynamics through task clustering and expert routing. We present the reconstructed images decoded by the final imagination states of the 26 tasks in Atari 100K, where the horizon is set to 16 steps. Compared to the vanilla transformer employed in vanilla STORM framework, the MoW demonstrates improved task discrimination and more accurate imagination in multi-task settings.

where the maximum operator is used to clip the loss below the value of $1$ nat $\approx 1.44$ bits (Chen et al., 2022) and $\mathrm{sg}(\cdot)$ is the stop-gradient operator.

To effectively alleviate the problem of gradient conflicts in multi-task learning and mitigate the scale disparities among loss terms, we employ harmonious loss weights to train the MoW.

### 3.4 HARMONIOUS AND BALANCED LOSS

The central challenges in MTRL lie in two key aspects: i) The first is the presence of gradient conflicts across tasks, which drive the shared parameters in divergent directions. This issue is further exacerbated in the end-to-end training pipelines involving multiple components, often resulting in degraded representations and unstable training dynamics (Chen et al., 2018). While prior works have introduced techniques such as gradient projection and conflict-aware optimization to mitigate this problem, these approaches typically incur additional model complexity. ii) The second challenge concerns the sensitivity of performance to the relative weighting of task-specific losses. Manually tuning these weights is both non-trivial and resource-intensive, thereby limiting the practicality of MTRL in real-world scenarios (Kendall et al., 2018).

To address these challenges, the MoW is trained end-to-end by incorporating the harmonious weights (Ma et al., 2024), which is applied on top of the individual task-specific losses (defined in Equation 9). This loss dynamically adjusts task weights to mitigate potential gradient conflicts during training, defined as follows

$$\mathcal{L}_{\mathcal{H}}(\phi) = \sum_{k=1}^{K} \frac{1}{\sigma_k} \mathcal{L}_k(\phi) + \ln(1 + \sigma_k), \tag{9}$$

where $\mathcal{L}_{\mathcal{H}}(\phi)$ is the harmonized loss and the rectified scale is equal to $\frac{2}{1+\sqrt{1+4/\mathbb{E}[\mathcal{L}_k]}} < 1$ (derived in Appendix A.7). Finally, the balance loss is added to the rectified loss

$$\mathcal{L}(\phi) = \mathcal{L}_{\mathcal{H}}(\phi) + 0.1\mathcal{L}_{\mathrm{bal}}(\phi), \tag{10}$$

with $\mathcal{L}_{\mathrm{bal}}(\phi) = \|\sum_{k=1}^{K} W_k - \frac{KN_k}{N_e}\mathbf{1}_{N_e}\|_2^2$, where $W_k$ is the normalized weight vector and $\mathbf{1}_{N_e}$ is a vector consisting of ones. Since the MoW does not explicitly apply experts' weights but instead combine expert features via concatenation, the balance loss primarily serves to encourage the activation of all experts, rather than enforcing identical expert selection distributions across tasks.

### 3.5 AGENT LEARNING

The agent learns entirely through the imagination trajectories enabled by the world model, making accurate reconstruction of the raw observations a critical factor for training efficient agents. As shown in Fig. 2, compared to the multi-task vanilla STORM, the MoW produces imagined trajectories that reconstruct the raw images more faithfully while avoiding confusion across tasks. We provide detailed reconstruction results of MoW across different tasks in Appendix A.6.

To initiate the imagination process, a short context trajectory is randomly sampled from the replay buffer to compute the initial posterior distribution $\mathcal{Z}_k^t$. During inference, rather than drawing samples from the posterior, the latent representation $z_k^t$ is sampled from the prior distribution $\hat{\mathcal{Z}}_k^t$. To accelerate inference, we incorporate the key-value (KV) caching mechanism within the Transformer architecture (Chen, 2022), which allows for faster autoregressive rollout by reusing previously computed attention keys and values. The agent's state is formed by concatenating $z_k^t$, $h_k^t$, and $e_k$

$$\text{State:} \qquad s_k^t = [z_k^t, h_k^t, e_k],$$

$$\text{Critic:} \quad V_{\psi,i_k}(s_k^t) \approx \mathbb{E}_{\pi_\theta,k}\Big[\sum_{\tau=0}^{\infty} \gamma_k^\tau r_k^{t+\tau}\Big], \tag{11}$$

$$\text{Actor:} \qquad a_k^t \sim \pi_\theta(a_k^t \mid s_k^t).$$

Other core training configurations follow the settings established in the STORM framework, for all tasks, the agent shares the actor network, while the number of critic networks matches the number of VAEs $N_m$. For task $T_k$, the corresponding critic network index is also determined through clustering the gradient vector during the warm-up stage. The complete loss of the actor-critic algorithm is described in Equation 12 and Equation 14, where $\hat{r}_k^t$ corresponds to the predicted reward, and $\hat{c}_k^t$ represents the predicted continuation flag. The critic networks learn to approximate the distribution of the $\lambda$-return estimates $R_{t,k}^\lambda$ using the maximum likelihood loss

$$\mathcal{L}(\psi) = \sum_{k,t} \Big[ \mathcal{L}_{\text{sym}}\big(V_{\psi,i_k}(s_k^t) - \text{sg}(R_{t,k}^\lambda)\big) + \mathcal{L}_{\text{sym}}\big(V_{\psi,i_k}(s_k^t) - \text{sg}(V_{\psi^{\text{EMA}},i_k}(s_k^t))\big) \Big], \tag{12}$$

where $\mathcal{L}_{\text{sym}}$ represents the symlog two-hot loss and $V_{\psi^{\text{EMA}}}$ is the exponential moving average of the critic network $\psi$ for stabilizing training and preventing overfitting. The $\lambda$-return $R_{t,k}^\lambda$ (Sutton & Barto, 2018; Hafner et al., 2025) is recursively defined as follows

$$R_{t,k}^\lambda = r_k^t + \lambda c_k^t\Big[(1-\lambda)V_{\psi,i}(s_k^{t+1}) + \lambda R_{t+1,k}^\lambda\Big],$$
$$R_{L,k}^\lambda = V_{\psi,i}(s_k^L). \tag{13}$$

The actor network is trained using the surrogate loss function

$$\mathcal{L}(\theta) = \sum_{k,t} \Big[ -\text{sg}\bigg(\frac{R_{t,k}^\lambda - V_{\psi,i}(s_k^t)}{\max(1,S)}\bigg) \ln \pi_\theta(a_k^t|s_k^t) - \eta H(\pi_\theta(a_k^t|s_k^t)) \Big], \tag{14}$$

where $H(\cdot)$ denotes the entropy of the policy distribution, and the constant $\eta$ represents the coefficient for entropy loss. The normalization ratio $S$ is defined in Equation 15, computed as the range between the 95 and 5 percentiles of the $\lambda$-return $R_{t,k}^\lambda$ across batch

$$S = \text{percentile}(R_{t,k}^\lambda, 95) - \text{percentile}(R_{t,k}^\lambda, 5). \tag{15}$$

## 3.6 WARMUPING MOW

World model-based RL methods typically perform a number of interactions with the environment using random actions prior to the world model training stage, populating the replay buffer to enable a stable initialization phase. During the warmup stage of MoW, after collecting a sufficient amount of random interaction data, we fix the replay buffer and refrain from updating the agent. Instead, we perform a number of world model training steps to initialize the dynamics representation. In this process, the expert Transformer modules are trained following the same end-to-end manner as previously described in Equation 10. However, only a single set of VAE and its associated predictors participate in the warmup training stage. Upon completion of the warm-up stage, the parameters of these components are copied to the remaining modules with identical architectures, after which the standard iterative updates of the world model and agent, as classic world model-based RL. After the warmup stage, we extract the gradient vector associated with each task. These task-specific gradient vectors are then clustered to determine which tasks should share the $i$-th set of VAE, predictor and critic network. This task grouping strategy based on the gradient correspondence is similar to HarmoDT (Hu et al., 2024). In practice, due to the large number of trainable parameters in our mixture world model, we reduce the computational cost by averaging the gradient vectors from each neural network layer. For a more comprehensive understanding of training the mixture world model, we provide the pseudo-code implementation in Appendix A.12.

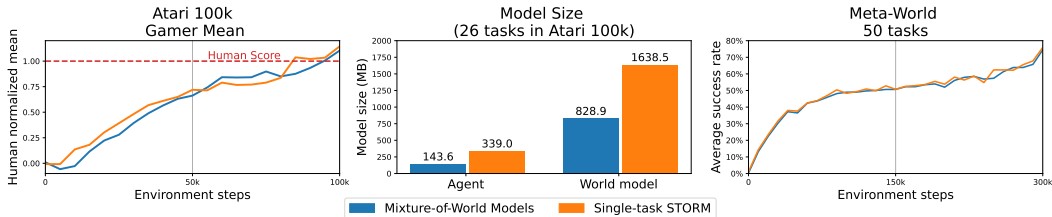

Figure 3: Results of MoW on the Atari 100K benchmark (left) and the Meta-world benchmark (right). Compared to baseline STORM, MoW results a $50\%$ reduction in model size (middle).

Table 1: Results on MetaWorld MT50

| Algorithms | Input | Success Rate (Episode Length) | Total Environment Steps |
|---|---|---|---|
| MTSAC |  | $49.3 \pm 1.5$ (150) |  |
| CARE | State | $50.8 \pm 1.0$ (150) | 20M |
| PaCo |  | $57.3 \pm 1.3$ (150) |  |
| MOORE | State | $72.9 \pm 3.3$ (150) | 100M |
| MoW (ours) | Image | $74.5 \pm 1.1$ (500) | 15M |

## 4 EXPERIMENTS

We evaluate the effectiveness of MoW in improving the sample efficiency of baseline MBRL methods across two representative benchmarks: Meta-world with robotic manipulation tasks (continuous action spaces) and Atari 100K with video games with discrete action spaces. Experimental results are shown in Figure 3 and demonstrate that our architecture not only substantially enhances sample efficiency but also achieves competitive performance with significantly fewer parameters compared to baselines. In our experiments, we use a machine with NVIDIA 4090 graphics cards with 8 CPU cores and 24 GB RAM. We use distributed data parallel (DDP) in Pytorch to enable multi-GPU parallel multi-task training, and training MoW on each Atari game for 100K steps took roughly 3.5 hours. Hyperparameters can be found in Appendix A.9.

### 4.1 ATARI 100K EXPERIMENTS

The Atari 100K benchmark is a widely adopted testbed for RL agents. To determine the performance of a human player $H$, the player is allowed to become familiar with the game under the same sample constraint. The MoW achieves a human normalized score of $110.4\%$ on the Atari 100K benchmark using a single unified model. To ensure the fairness and quality of our results, we also reproduced STORM results using the official code and configurations, which is a human normalized score of $114.2\%$. Furthermore, compared to STORM, our primary baseline, which requires $1,977.5$ MB to support 26 tasks, the MoW achieves comparable performance with a significantly more compact model of only $972.5$ MB. Here, MB refers to the storage size of the actual model checkpoint (.pth file), corresponding to a $50\%$ reduction in model size. Details are provided in Appendix A.10.1.

### 4.2 META-WORLD EXPERIMENTS

Meta-World is a benchmark of 50 robotic manipulation tasks with fine-grained observation details, such as small target objects. In all tasks, the episode length is 500 environment steps with no action repeat. In this benchmark, performance is primarily measured by the average success rate across all tasks, offering a holistic evaluation of the robot system's adaptability and competence in varied task settings. We compare MoW with the official results of state-of-the-art MTRL methods based on state inputs. As stated in Table 1, although MoW makes decisions solely from the visual observations-a considerably more challenging setting than state-based decision-making, it still achieves a success rate of $74.5\%$ within only 300K steps per task (15M steps in total). Notably, this surpasses the

performance of classic model-free baselines trained for 20M steps, highlighting MoW's remarkable sample efficiency.

### 4.3 PARAMETER SCALABILITY

Similar to the MoE architecture, MoW exhibits strong scalability to a large number of tasks, owing to its parameter-efficient design. Specifically, MoW enhances overall performance by increasing the number of expert modules, effectively reducing the task load per expert. As illustrated in Figure 4, MoW achieves competitive results across 26 tasks in the Atari 100K benchmark with only 12 experts. Within a reasonable range, scaling up the model size leads to substantial performance gains. Notably, increasing the number of expert Transformers yields more pronounced improvements, as these modules directly facilitate task-specific dynamics modeling. While adding additional VAEs also influences performance by improving reconstruction precision, their impact is comparatively modest when the number of expert Transformers is held constant.

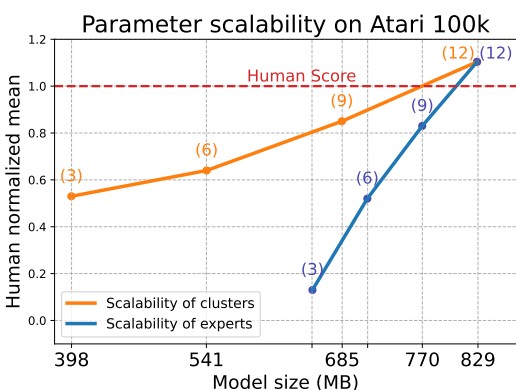

Figure 4: By increasing the number of experts (blue) and clusters of VAEs (yellow), MoW effectively unlocks the parameter scalability.

## 5 ABLATION STUDIES

To further investigate the advantages of MoW, we first extended the vanilla STORM framework to a multi-task setting by incorporating task embeddings. The resulting model failed to train effectively, indicating that the baseline is not readily adaptable to multi-task scenarios. To ensure this limitation was not due to the VAE, we introduced multiple VAEs into the vanilla STORM baseline. The results show that the vanilla Transformer configuration did not achieve performance comparable to our mixture architecture. As a result, the agent was unable to learn effectively within the latent space. In particular, it suffered from significantly higher dynamics modeling loss, suggesting that the imagination process was severely impaired (as shown in Figure 2).

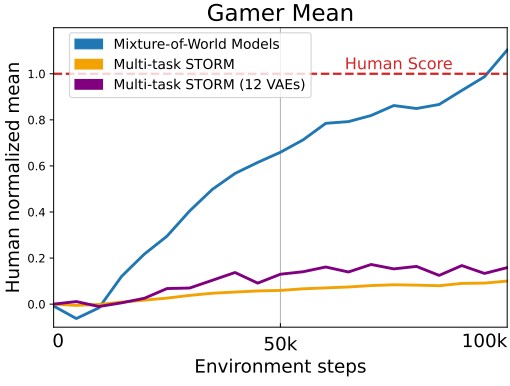

Figure 5: Ablation studies on muti-task STORM.

## 6 CONCLUSION

In this work, we introduced a MTRL approach based on the modular mixture-of-world models (MoW) architecture. By combining multiple categorical VAEs, parallel expert Transformers, and a shared Transformer, the MoW effectively captures diverse task dynamics while maintaining high sample efficiency. The use of task prediction and expert balance losses further enhances task discrimination and expert utilization, respectively. Additionally, the proposed warmup stage promotes efficient module sharing through gradient-based task clustering, substantially reducing model complexity. Substantial experiments on Atari 100K and Meta-World benchmarks demonstrate that the MoW not only surpasses prior world model baselines, but also achieves superior performance with significantly fewer parameters. These results underscore the potential of MoW as a scalable and efficient framework for general MTRL.

## ACKNOWLEDGMENTS

This work was supported in part by the National Natural Science Foundation of China under Grants 62495095, 62495090, and U23B2059.

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

# A    APPENDIX

## A.1    FURTHER ANALYSIS

We highlight the distinctions between MoW and recent approaches in the world model and MTRL as follows:

Table 2: Comparison between MoW and recent approaches.

| Attributes | CARE | MOORE | DreamerV3 | STORM | MoW (ours) |
|---|---|---|---|---|---|
| MBRL/MFRL | Model free | Model free | Model based | Model based | Model based |
| Setting | Multi-task | Multi-task | Single-task | Single-task | Multi-task |
| World Model | - | - | GRU | Transformer | Hybrid Transformer |
| Input | State | State | Image | Image | Image |
| Agent training | MTSAC | MTSAC | DreamerV3 | As DreamerV3 | Multi-task STORM |

- CARE (Sodhani et al., 2021a) and MOORE (Hendawy et al., 2023) are model-free multi-task RL methods that suffer from low sample efficiency, whereas DreamerV3 (Hafner et al., 2025) and STORM (Zhang et al., 2023) are world-model-based single-task reinforcement learning approaches that exhibit high sample efficiency but lack multi-task implementations. In contrast, MoW is a world model-based MTRL method that combines high sample efficiency with strong multi-task performance.

- In contrast to DreamerV3 (Hafner et al., 2025), which employs a GRU (Dey & Salem, 2017), and STORM (Zhang et al., 2023), which adopts a vanilla Transformer as the sequence model, MoW introduces a novel hybrid Transformer architecture that better captures diverse task dynamics while preserving reconstruction fidelity.

- CARE (Sodhani et al., 2021a) and MOORE (Hendawy et al., 2023) are state-based methods that build on the classical MTSAC algorithm for decision-making, with their primary focus on introducing MoE architectures for policy networks and task-specific training strategies. DreamerV3 (Hafner et al., 2025) and STORM (Zhang et al., 2023), in contrast, adopt actor–critic methods to train the agent on imagined trajectories, with their main contributions centered on encoding and predicting more accurate image observations by the hidden representations. MoW extends the architecture of STORM by leveraging gradient-based clustering to allocate distinct critic networks to different tasks, while still employing a similar actor–critic training paradigm for the agent.

## A.2    THE USE OF LARGE LANGUAGE MODELS (LLMS)

LLMs are employed to assist in detecting and rectifying potential grammatical errors in the manuscript.

### A.3 GRADIENT-BASED CLUSTERING

The warmup stage in MoW is a short offline world-model self-supervised training phase on a fixed replay buffer (as we shown in Algorithm 1). Before clustering, we first collect a sufficient amount of random interaction data for all tasks and then train a single VAE–predictor stack on this fixed dataset while keeping the agent frozen (Sec. 3.6). This stage corresponds to self-supervised learning of reconstruction, reward, continuation, dynamic and task prediction losses, rather than unstable online policy optimization. As we shown in the Figure 6, these losses converge within a few thousand optimization steps, indicating that the world model has already fitted the warmup dataset well. In our experiments, varying the warmup length within a reasonable range (e.g., 5000 steps) has only minor impact on final Atari and Meta-World performance, suggesting that MoW is not overly sensitive to the exact duration of warmup, as long as the losses have essentially stabilized.

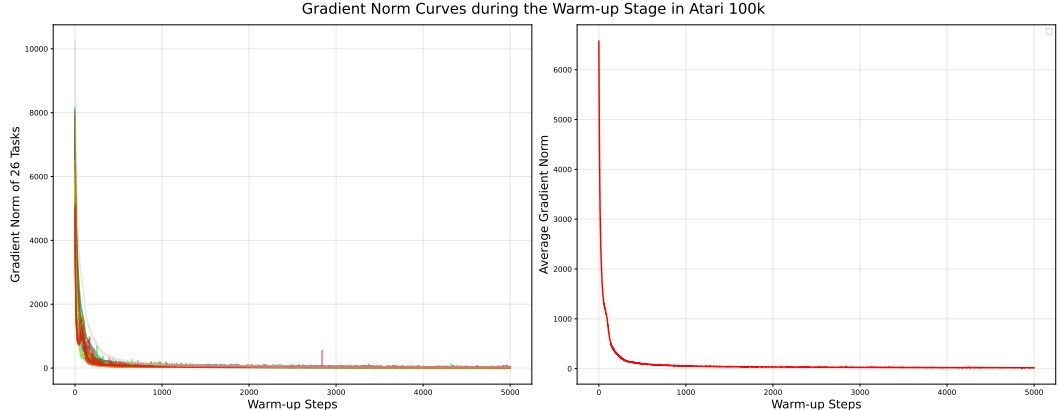

Figure 6: The gradient norm curves in the warmup stage of MoW.

## A.4 CHALLENGES IN HIGH-DIMENSIONAL OBSERVATIONS

Reinforcement learning in high-dimensional observation spaces, such as those encountered in visual tasks, presents several challenges compared to state-based RL, where observations are typically simpler and lower-dimensional. These challenges arise due to the complexities of learning and processing high-dimensional data, such as images or videos, and the need for efficient generalization across multiple tasks. In state-based RL, the observation space is often relatively small (e.g., a $39-$dimension vector representing the agent's position and velocity (Sodhani et al., 2021b)), and traditional MTRL algorithms can work effectively by directly modeling these state representations. In contrast, high-dimensional observations, such as images, have large input spaces and complex structures that must be compressed into useful representations. In addition, state-based RL typically utilizes a single, simpler model, as state representations are usually more structured and lower-dimensional. However, high-dimensional visual inputs require specialized visual modules, and the online optimization of these modules during training imposes a significant burden on the convergence of reinforcement learning.

To further illustrate the challenges of high-dimensional visual MTRL compared to state-based MTRL, we use official results from the classic multi-task method TDMPC2 (Hansen et al., 2024), and the results is shown in Figure 7. In the 12 tasks of Deepmind Control, the performance of TDMPC2 with visual pixels inputs (the orange lines) is worse and converges more slowly compared to the state-based inputs (the blue lines). It fails to converge after 2M steps of single-task interaction. To ensure fairness, the only difference between these two models is the addition of a shallow convolutional encoder in the visual input version, with all other parameters being identical.

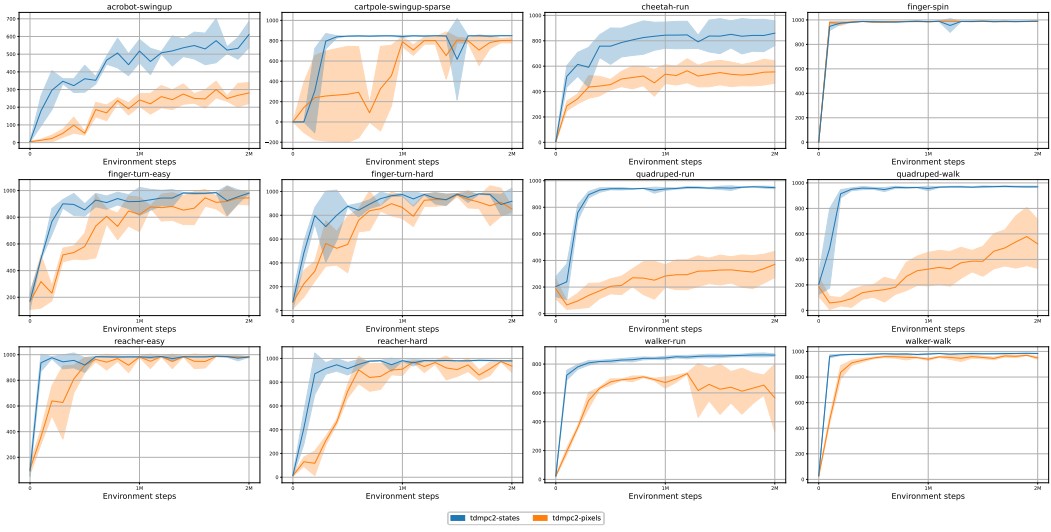

Figure 7: Results of pixels-input and states-input TDMPC2 on DMC suite.

## A.5 Additional Ablation Studies

Due to the page limitations in the main text, we provide additional ablation on all of our design choices for MoW in this section, including the clusters of VAE-predictors, number of experts, cascaded expert-shared transformers, task predictor head, balanced loss, and the gradient-based task clustering. The results are shown in Figure 8 and 9.

**Clusters of VAE-predictors.** We ablate the number of clusters of VAE-predictors $N_m$ from 12 to 3 and observe a clear performance degradation as the clusters count decreases. With fewer clusters, each VAE-predictor must capture the visual characteristics of a larger set of tasks, which slows down convergence and leads to lower overall performance.

**Number of Expert Transformers.** We ablate the number of expert Transformers $N_e$ from 12 to 3 (when $N_e = 1$, the model degenerates into a single shared Transformer) and observe a clear performance degradation as the experts count decreases. With fewer experts, it becomes more difficult to capture the diverse dynamics across tasks, which reduces the quality of imagined trajectories generated by the world model and ultimately makes policy learning more challenging.

**Expert and Shared Transformers.** We ablate the shared Transformer and observe a modest performance drop, as tasks in Atari 100K share relatively little inter-task structure. In contrast, removing the expert Transformer leads to a much larger degradation, since the expert Transformer is responsible for capturing task-specific dynamics and generating accurate multi-task imagined trajectories.

**Task Prediction.** We ablate the task predictor by removing both the prediction head and its associated loss. The task predictor plays an important role in maintaining task-discriminative structure in the learned latent space, which benefits both world-model learning and downstream RL performance. Consistent with this, removing the task predictor leads to a noticeable decline in overall performance.

**Harmonious Loss.** We ablate the harmonious loss by removing the harmonious weight in Equation 9. Without this operation, the world-model training faces the gradient conflicting problem, and the end-to-end loss is dominated by tasks with larger loss magnitudes, leading to unstable training and reduce the RL performance. This confirms that harmonizing the multi-task losses is important for stable and effective multi-task visual world-model learning.

**Balance Loss.** To evaluate whether the expert-balance term is necessary to prevent overuse of a few experts, we remove the balance loss and observe that the training becomes sensitive to initialization. The balance loss is critical for ensuring all experts are utilized and prevents collapse into degenerate expert usage.

**Gradient-based Task Clustering.** We use randomly partition tasks into the same number of clusters ($N_m = 12$) to ablate the gradient-based warmup clustering. The results show that the gradient-based clustering is necessary for allocating shared parameters meaningfully and avoiding interference across unrelated tasks.

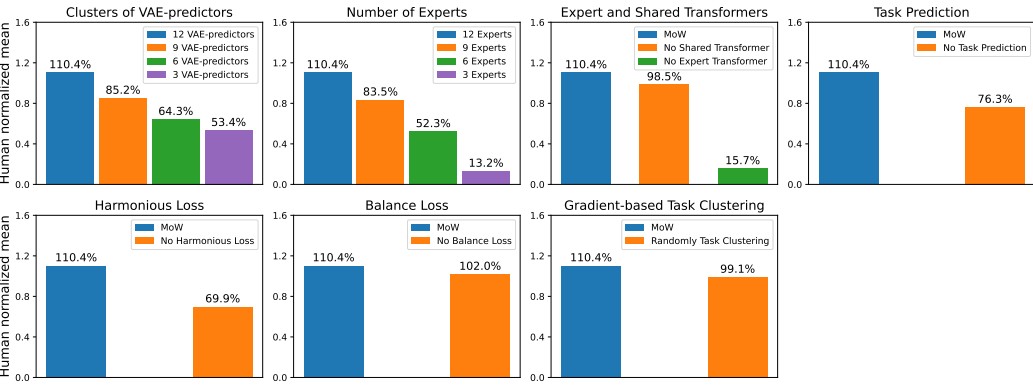

Figure 8: Results of additional ablation studies

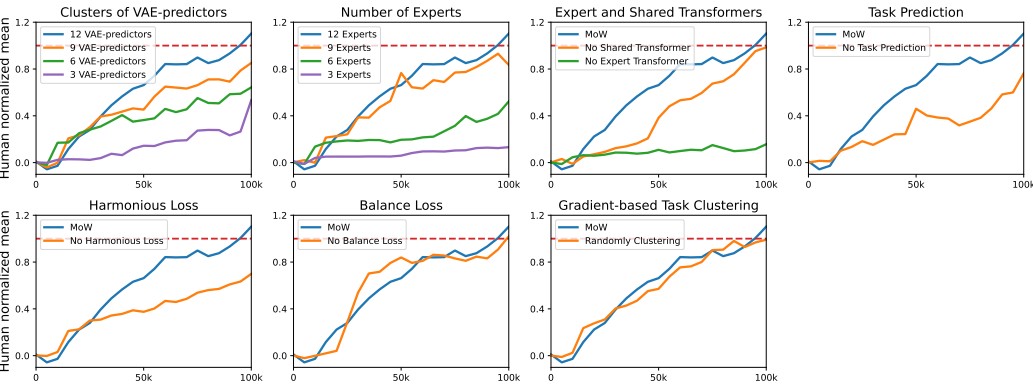

Figure 9: Results of additional ablation studies

## A.6 RECONSTRUCTION BY IMAGINATION

We provide Multi-step predictions on several environments in Atari games. MoW utilizes 8 observations and actions as contextual input, enabling the imagination of future events in an auto-regressive manner.

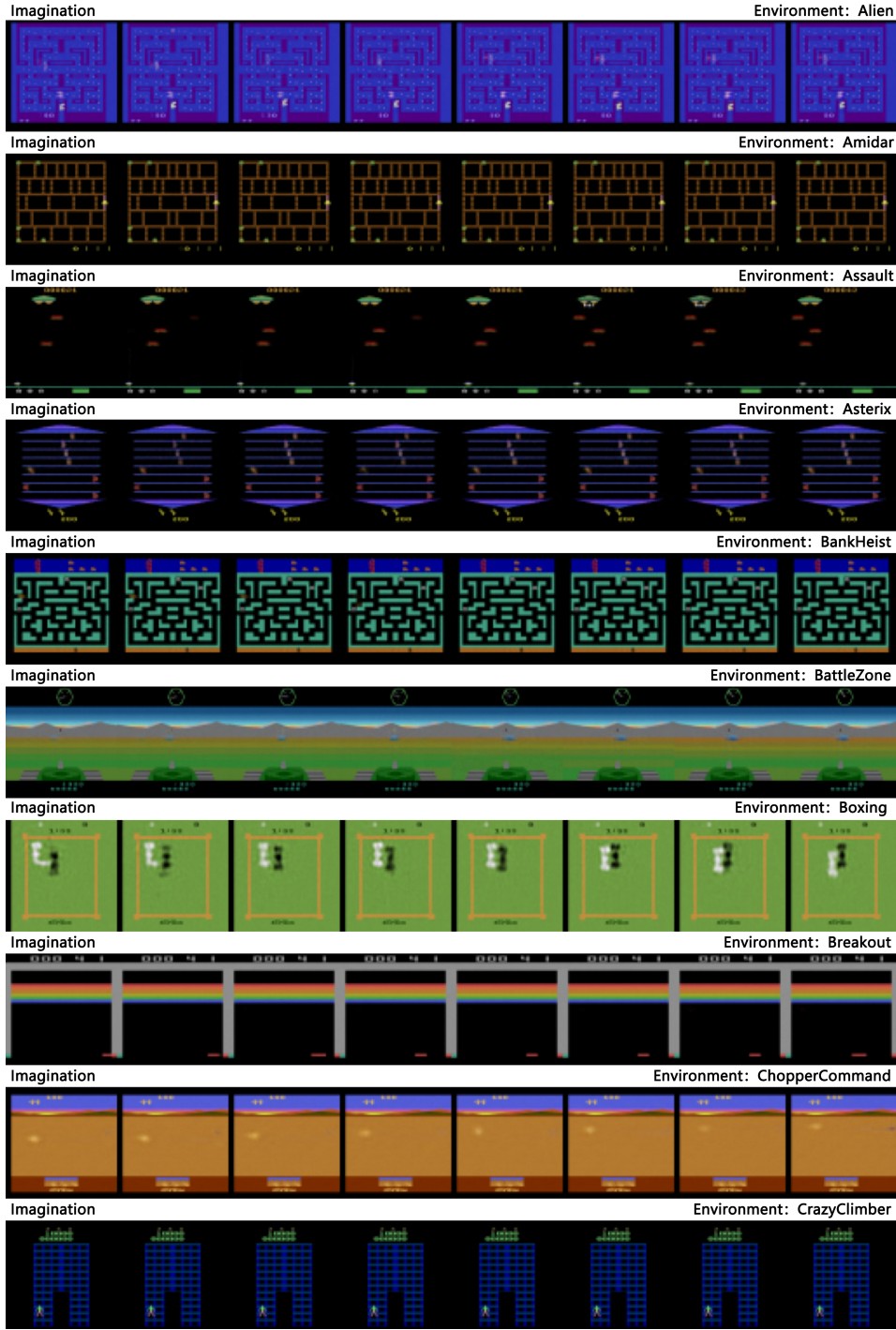

Figure 10: Multi-step predictions on several environments in Atari games.

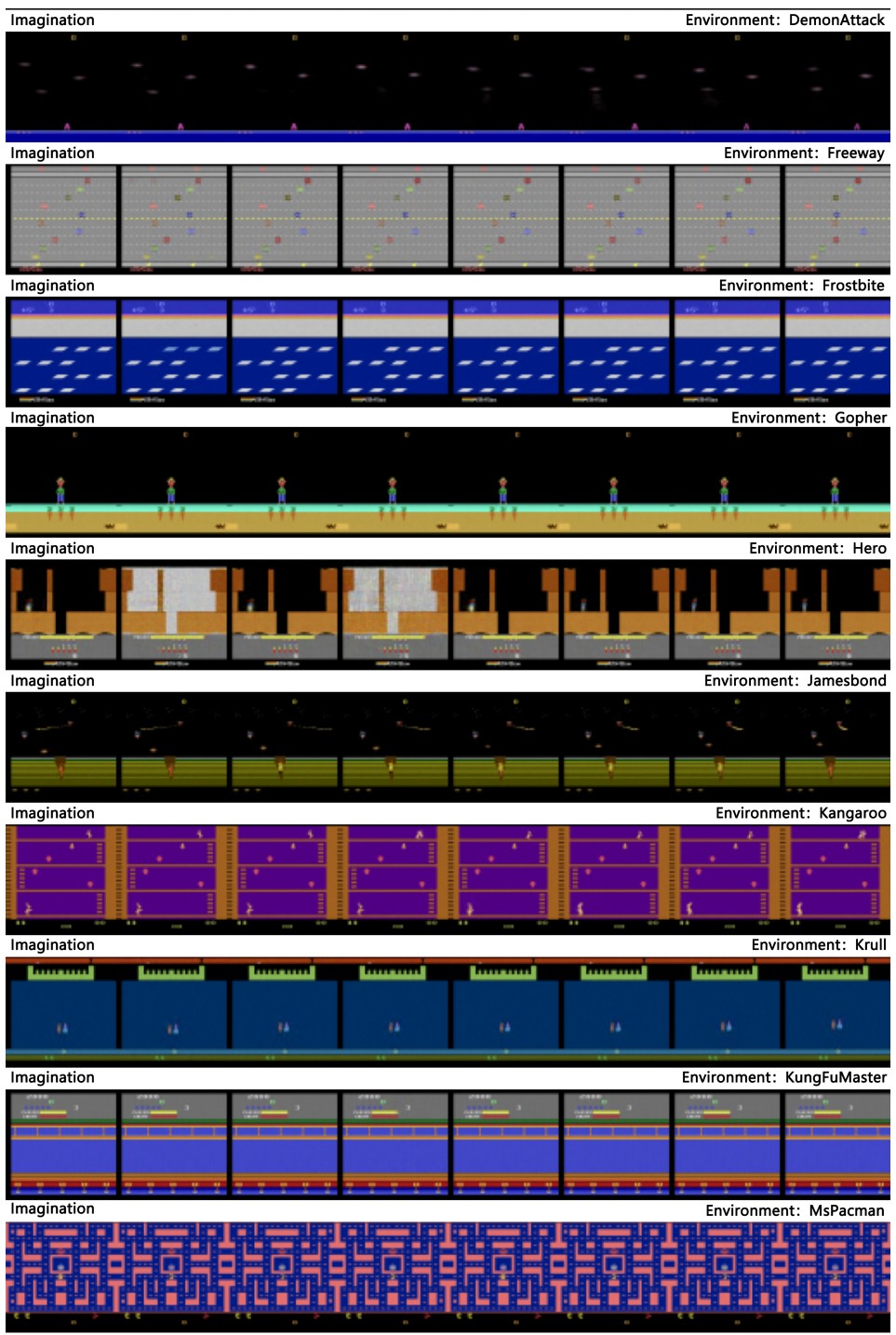

Figure 11: Multi-step predictions on several environments in Atari games.

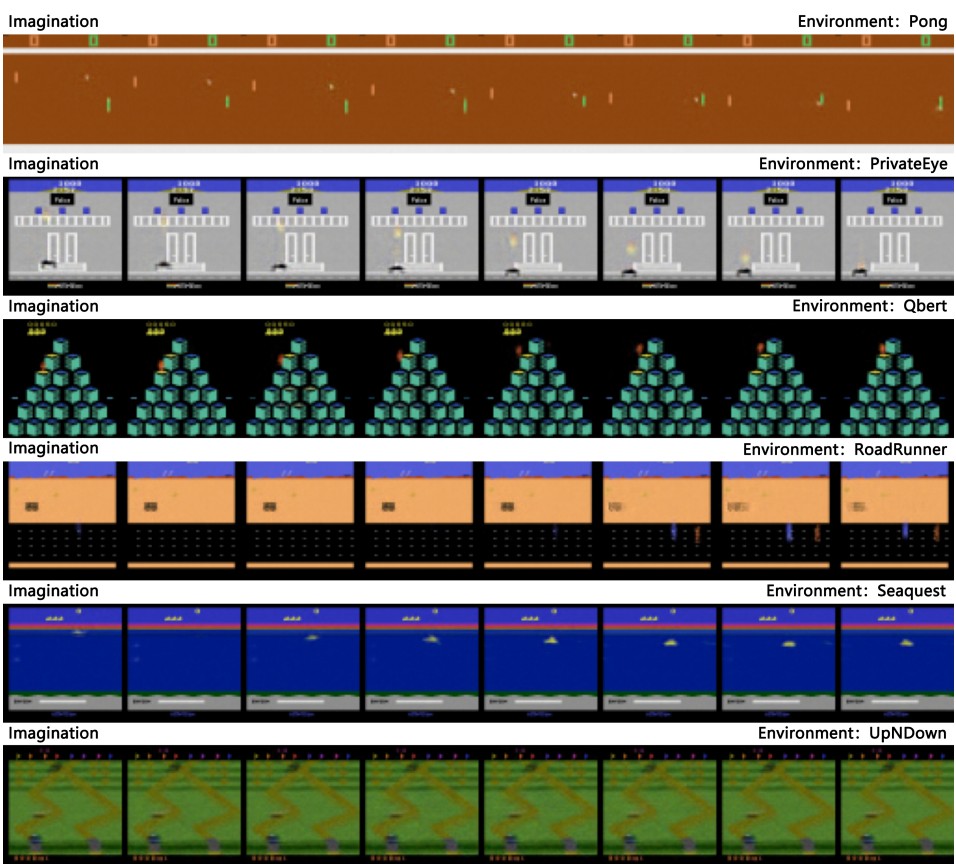

Figure 12: Multi-step predictions on several environments in Atari games.

## A.7 HARMONIOUS LOSS

Adjusting the coefficients of different loss terms has the potential to significantly enhance the performance of world model-based reinforcement learning algorithms. The harmonious loss strategy dynamically adjusts the dominance between the observation and the reward modeling during world model training process. The harmonious loss is constructed as:

$$\mathcal{L}_{\mathcal{H}}(\phi) = \sum_{k=1}^{K} \frac{1}{\sigma_k} \mathcal{L}_k(\phi) + \ln(1 + \sigma_k). \tag{16}$$

The reason why we use $\ln(1 + \sigma_k)$ as regularization terms is that $\sigma_k$ is parameterized as $\sigma_k = \exp(\sigma_k) > 0$ to optimize parameters $\sigma_k$ free of sign constraint. Since a loss item with small values, such as the reward loss, can lead to extremely large coefficient $1/\sigma_k \approx \mathcal{L}_k - 1 \gg 1$, which potentially hurt training stability. The derivation of the harmonious loss scale is as follows.

To minimize $\mathbb{E}[\mathcal{L}_{\mathcal{H}}(\phi, \sigma_k)]$, we force the the partial derivative w.r.t. $\sigma_k$ to 0:

$$\nabla_\sigma \mathbb{E}[\mathcal{L}_{\mathcal{H}}(\phi, \sigma)] = \nabla_\sigma \left( \frac{1}{\sigma_k} \mathbb{E}[\mathcal{L}_k] + \ln(1 + \sigma_k) \right) = -\frac{1}{\sigma^2} \mathbb{E}[\mathcal{L}_k] + \frac{1}{1 + \sigma_k}$$
$$\sigma_k^* = \frac{\mathbb{E}[\mathcal{L}_k] + \sqrt{\mathbb{E}[\mathcal{L}_k]^2 + 4\mathbb{E}[\mathcal{L}_k]}}{2} \tag{17}$$

Therefore the learnable loss weight, in the rectified harmonious loss, approximates the analytic loss weight:

$$\frac{1}{\sigma_k^*} = \frac{2}{\mathbb{E}[\mathcal{L}_k] + \sqrt{\mathbb{E}[\mathcal{L}_k]^2 + 4\mathbb{E}[\mathcal{L}_k]}} \tag{18}$$

and equivalently, the harmonized loss scale is:

$$\mathbb{E}[\frac{\mathcal{L}_k}{\sigma_k^*}] = \frac{2}{1 + \sqrt{1 + \frac{4}{\mathbb{E}[\mathcal{L}_k]}}} < 1, \tag{19}$$

and add the regularization term $\ln(1 + \sigma_k)$ results in the $4/\mathbb{E}[\mathcal{L}_k]$ in the $\sqrt{1 + 4/\mathbb{E}[\mathcal{L}_k]}$ term, which prevents the loss weight from getting extremely large when faced with a small $\mathbb{E}[\mathcal{L}_k]$.

A.8 DETAILS OF MODEL STRUCTURE

Table 3 and Table 4 show the structure of the image encoder and decoder. The sizes of the submodules are omitted as they can be directly inferred from the tensor shapes. ReLU denotes the rectified linear unit activation, and Linear corresponds to a fully connected layer. Flatten and Reshape operations are applied solely to modify the indexing of tensors while preserving both the data and their original ordering. Conv refers to a CNN layer(LeCun et al., 1989), characterized by kernel $= 4$, stride $= 2$, and padding $= 1$. DeConv denotes a transpose CNN layer (Zeiler et al., 2010), also characterized by kernel $= 4$, stride $= 2$, and padding $= 1$. BN denotes the batch normalization layer (Ioffe & Szegedy, 2015). The variable $E$ represents the dimension of task embedding $e_k$.

Table 3: Structure of the image encoder.

| Submodule | Output tensor shape |
|---|---|
| Input image ($o_k^t$) | $3 \times 64 \times 64$ |
| Conv1 + BN1 + ReLU | $32 \times 32 \times 32$ |
| Conv2 + BN2 + ReLU | $64 \times 16 \times 16$ |
| Conv3 + BN3 + ReLU | $128 \times 8 \times 8$ |
| Conv4 + BN4 + ReLU | $256 \times 4 \times 4$ |
| Flatten | 4096 |
| Linear | 1024 |

Table 4: Structure of the image decoder.

| Submodule | Output tensor shape |
|---|---|
| Random sample ($z_k^t$) | $32 \times 32$ |
| Flatten and task embedded | $1024 + E$ |
| Linear+BN0+ReLU | 4096 |
| Reshape | $256 \times 4 \times 4$ |
| DeConv1 + BN1 + ReLU | $128 \times 8 \times 8$ |
| DeConv2 + BN2 + ReLU | $64 \times 16 \times 16$ |
| DeConv3 + BN3 + ReLU | $32 \times 32 \times 32$ |
| DeConv4 (produce $o_k^t$) | $3 \times 64 \times 64$ |

Table 5: Action mixer $m_{j,k}^{1:t} = m_{\phi,j}(z_k^{1:t}, a_k^{1:t}, e_k)$. Concatenate denotes combining the last dimension of tensors and merging them into one new tensor. The variable $A$ represents the action dimension, which is 18 for Atari and 4 for Meta-world. $D$ denotes the feature dimension of the Transformer. LN is an abbreviation for layer normalization

| Submodule | Output tensor shape |
|---|---|
| Random sample ($z_k^t$), action ($a_k^t$), task embedding ($e_k$) | $32 \times 32, A, E$ |
| Reshape and concatenate | $1024 + E + A$ |
| Linear1+LN1+ReLU | $D$ |
| Linear2+LN2+ReLU | $D$ |

Table 6: Transformer block. Dropout (Vaswani, 2017) is used in each Transformer submodule to reduce overfitting. We also apply Dropout to attention weights in the MHSA module.

| Submodule | Module alias | Output tensor shape |
|---|---|---|
| Input features (label as $x_1$) | | $32 \times 32$ |
| Multi-head self attention
Linear1 + Dropout
Residual (add $x_1$)
LN1(label as $x_2$) | MHSA | $T \times D$ |
| Linear2 + ReLU
Linear3 + Dropout
Residual (add $x_2$)
LN2 | FFN | $T \times 2D$
$T \times D$
$T \times D$
$T \times D$ |

Table 7: Expert Transformer $l_{j,k}^{1:t} = f_{\phi,j}(m_{j,k}^{1:t})$, where $j$ is the index of the expert.

| Submodule | Output tensor shape |
|---|---|
| Input ($m_{j,k}^{1:t}$) | $T \times D$ |
| Transformer blocks $\times M$ | $T \times D$ |
| Output ($l_{j,k}^{1:t}$) | $T \times D$ |

Table 8: Shared Transformer $h_k^{1:t} = F_\phi(l_k^{1:t}, e_k)$.

| Submodule | Output tensor shape |
|---|---|
| Input ($l_k^{1:t}, e_k$) | $T \times (n_k \times D + E)$ |
| Reshape | $T \times D$ |
| Transformer blocks $\times M$ | $T \times D$ |
| Output ($h_k^{1:t}$) | $T \times D$ |

Table 9: Pure MLP structures. A $1-$layer MLP corresponds to a fully-connected layer, 255 is the size of the bucket of symlog two-hot loss, and $K$ is the number of tasks.

| Submodule | MLP layers | Input/ MLP hidden/ Output dimension |
|---|---|---|
| Task embedding ($e_k$) | 1 | $1/-/96$ |
| Dynamic predictor ($g_{\phi,i_k}^D$) | 1 | $D+E/-/1024$ |
| Reward predictor ($g_{\phi,i_k}^R$) | 3 | $D/D/255$ |
| Continuation predictor ($g_{\phi,i_k}^C$) | 3 | $D/D/1$ |
| Task predictor ($g_\phi^T$) | 1 | $D/-/K$ |
| Policy network ($V_{\psi,i_k}$) | 3 | $D/D/A$ |
| Actor network ($\pi_\theta(a_k^t \mid s_k^t)$) | 3 | $D/D/255$ |

## A.9 HYPERPARAMETERS

Table 10 detail hyperparameters of the optimization and environment, as well as hyperparameters shared by multiple components. Hyerparameters.

Table 10: Hyperparameters

| Name | Value |
| --- | --- |
| World model training batch size | 16 |
| World model training batch length | 64 |
| Imagination batch size | 1024 |
| Imagination context length | 8 |
| Imagination horizon | 16 |
| Warmup steps | 5000 |
| Environment context length | 16(Atari100K) /32(Meta-world) |
| Update world model every env step | 1 |
| Update agent model every env step | 1 |
| Number of experts | 12(Atari100K) /20(Meta-world) |
| Number of activated experts | 3(Atari100K) /4(Meta-world) |
| Number of VAE clusters | 12(Atari100K) /20(Meta-world) |
| Learning rate | $10^{-4}$ |
| World model gradient clipping | 1000 |
| Adam epsilon | $10^{-8}$ |
| Optimizer | Adam |
| Expert Transformer layers | 2 |
| Shared Transformer layers | 2 |
| Expert Transformer heads | 8 |
| Shared Transformer heads | 8 |
| Dropout | 0.1 |
| Discount horizon | 333 |
| Return lambda | 0.95 |
| Critic EMA decay | 0.98 |
| Critic EMA regularizer | 1 |
| Actor entropy scale | $3 \times 10^{-4}$ |
| Learning rate | $3 \times 10^{-5}$ |
| Actor-critic gradient clipping | 100 |

## A.10   EXPERIMENTAL DETAILS

### A.10.1   ATARI 100K CURVES AND AGGREGATE SCORES

We evaluate agent performance over 100 episodes using the final model checkpoint, saved every 2500 steps.

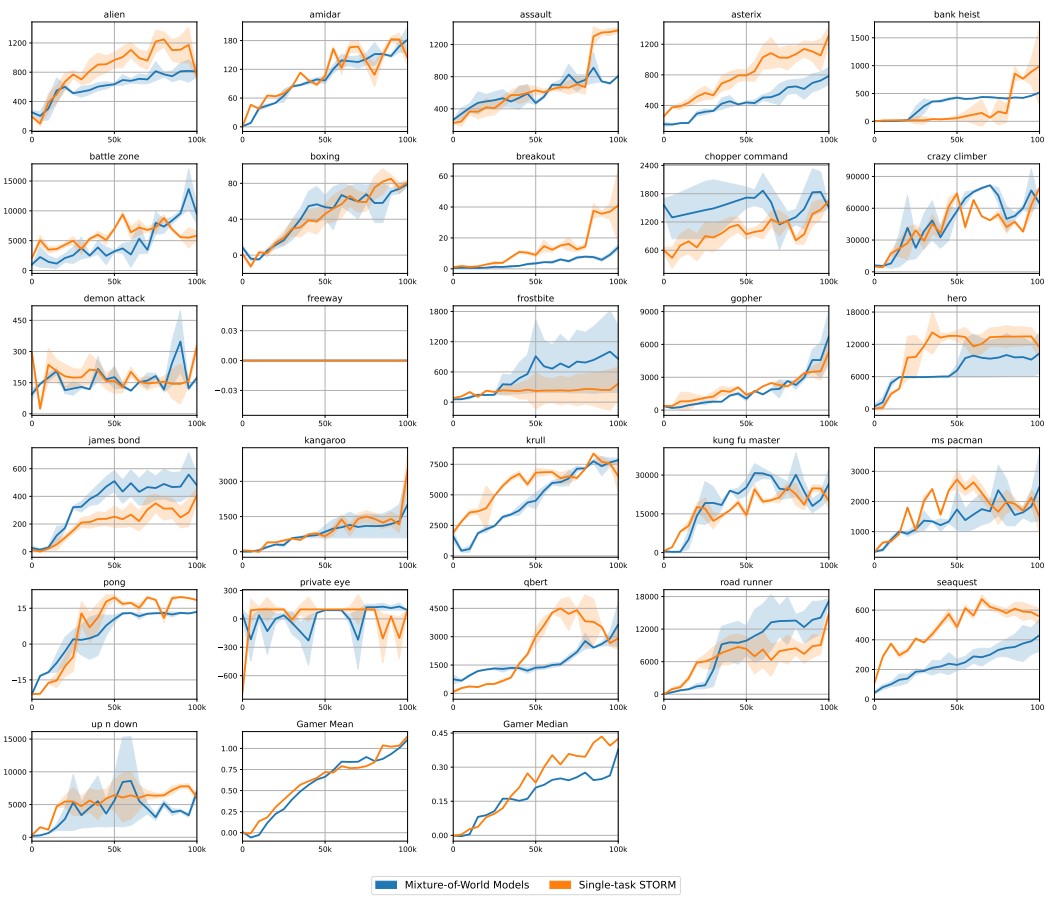

Figure 13:   Evaluation curves of MoW and single-task STORM on the Atari100K benchmark for individual games (400K environment steps). The solid lines represent the average scores over 5 seeds, and the filled areas indicate the standard deviation across these 5 seeds.

Table 11 summarizes the aggregate scores of MoW and some Transformer-based world model baselines (Robine et al., 2023; Micheli et al., 2022).

### A.10.2   META-WORLD

The Meta-World benchmark encompasses a diverse array of 50 distinct manipulation tasks, unified by shared dynamics. These tasks involve a Sawyer robot engaging with a variety of objects, each distinguished by unique shapes, joints, and connective properties. The complexity of this benchmark lies in the heterogeneity of the observation spaces and reward functions across tasks, as the robot is required to manipulate different objects towards varying objectives. The robot operates with a 4-dimensional fine-grained action input at each timestep, which controls the 3D positional movements of its end effector and modulates the gripper's openness. In order to use a single camera viewpoint consistently over all 50 tasks, we use the modified corner2 camera viewpoint for all tasks. Specifically, we adjusted the camera position with env.model.cam pos[2][:]=[0.75, 0.075, 0.7], which enables us to solve non-zero success rate on all tasks. Maximum episode length for

Table 11: Game scores and overall human-normalized performance on the 26 games in the Atari 100K benchmark.

| Game | Random | Human | STORM (original) | STORM (reproduced) | MoW (ours) | TWM | IRIS | EfficientZero | BBF | EfficientZeroV2 |
|---|---|---|---|---|---|---|---|---|---|---|
| Setting | | | Single-task | Single-task | **Multi-task** | Single-task | Single-task | Single-task | Single-task | Single-task |
| Alien | 228 | 7128 | 984 | 748 | 811 | 675 | 420 | 809 | 1173 | 1558 |
| Amidar | 6 | 1720 | 205 | 144 | 182 | 122 | 143 | 149 | 245 | 185 |
| Assault | 222 | 742 | 801 | 1377 | 808 | 683 | 1524 | 1263 | 2099 | 1758 |
| Asterix | 210 | 8503 | 1028 | 1318 | 1319 | 1116 | 854 | 25558 | 3946 | 61810 |
| Bank Heist | 14 | 753 | 641 | 990 | 517 | 467 | 53 | 351 | 733 | 1317 |
| Battle Zone | 2360 | 37188 | 13540 | 5830 | 9460 | 5068 | 13871 | 13871 | 24460 | 14433 |
| Boxing | 0 | 12 | 80 | 81 | 79 | 78 | 70 | 53 | 86 | 75 |
| Breakout | 2 | 30 | 16 | 41 | 14 | 20 | 84 | 414 | 371 | 400 |
| Chopper Command | 811 | 7388 | 1888 | 1644 | 1504 | 1697 | 1565 | 1117 | 7549 | 1197 |
| Crazy Climber | 10780 | 35829 | 66776 | 79196 | 64554 | 71820 | 59234 | 83940 | 58432 | 112363 |
| Demon Attack | 152 | 1971 | 165 | 324 | 173 | 350 | 2034 | 13004 | 13341 | 22773 |
| Freeway | 0 | 30 | 0 | 0 | 0 | 24 | 31 | 22 | 26 | 0 |
| Frostbite | 65 | 4335 | 1316 | 366 | 859 | 1476 | 259 | 296 | 2385 | 1136 |
| Gopher | 258 | 2413 | 8240 | 5307 | 6742 | 1675 | 2236 | 3260 | 1331 | 3869 |
| Hero | 1027 | 30826 | 11044 | 11434 | 10339 | 7254 | 7037 | 9316 | 7819 | 9705 |
| James Bond | 29 | 303 | 509 | 408 | 480 | 362 | 463 | 517 | 1130 | 468 |
| Kangaroo | 52 | 3035 | 4208 | 3512 | 2014 | 1240 | 838 | 724 | 6615 | 1887 |
| Krull | 1598 | 2666 | 8413 | 6522 | 7841 | 6349 | 6616 | 5663 | 8223 | 9080 |
| Kung Fu Master | 256 | 22736 | 26182 | 20046 | 26598 | 24555 | 21760 | 30945 | 18992 | 28883 |
| Ms Pacman | 307 | 6952 | 2673 | 1490 | 2492 | 1588 | 999 | 1281 | 2008 | 2251 |
| Pong | -21 | 15 | 11 | 18 | 13 | 19 | 15 | 20 | 17 | 21 |
| Private Eye | 25 | 69571 | 7781 | 100 | 92 | 87 | 100 | 97 | 41 | 100 |
| Qbert | 164 | 13455 | 4522 | 2910 | 3641 | 3331 | 746 | 13782 | 4447 | 16058 |
| Road Runner | 12 | 7845 | 17564 | 14841 | 17169 | 9109 | 9615 | 17751 | 33427 | 27516 |
| Seaquest | 68 | 42055 | 525 | 557 | 431 | 774 | 661 | 1100 | 1233 | 1974 |
| Up N Down | 533 | 11693 | 7985 | 6128 | 6943 | 15982 | 3546 | 17264 | 12102 | 15224 |
| Human Mean | 0% | 100% | 126.7% | 114.3% | 110.4% | 96% | 105% | 194.5% | 224.7% | 242.8% |
| Human Median | 0% | 100% | 58.4% | 42.5% | 37.7% | 51% | 29% | 109.0% | 91.7% | 128.6% |

Meta-world tasks is 500. We follow the environment setup of Meta-World as specified in Masked World Model (MWM) (Seo et al., 2023), with the exception of using a different action repeat setting (2 in MWM but 1 in MoW). All curves are shown in Figure 14.

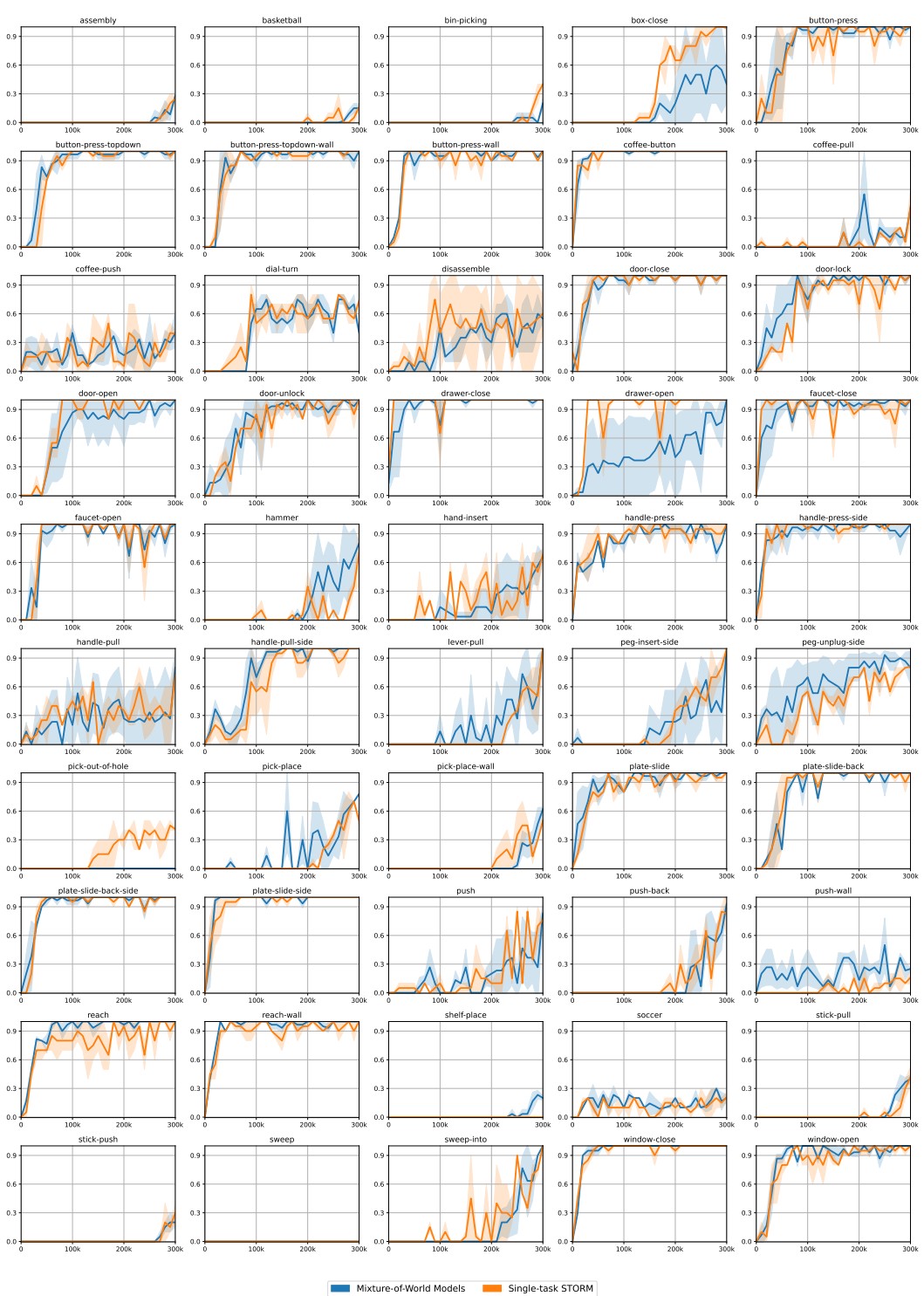

Figure 14: Evaluation curves of MoW on the Meta-World benchmark.

### A.11 BENCHMARKING TD-MPC2

We compare the performance of MoW with TD-MPC2 on the Meta-World benchmark. In the official implementation of TD-MPC2, the multi-task learning setup is based on large offline datasets (545M transitions), where the model is trained on the fixed state-based data rather than learning directly from online interactions. Moreover, TD-MPC2 is designed for state-based inputs, unlike MoW, which is tailored to handle high-dimensional visual inputs. To make a fair comparison in the context of visual reinforcement learning (RL), we adapted the official state-based implementation of TD-MPC2 by incorporating a shallow convolutional network to handle visual inputs (this operation is same to the visual RL experiments in TD-MPC2). We then tested this modified version of TD-MPC2 against MoW in the same Meta-World tasks. This modification allowed us to assess the performance of TD-MPC2 in a visual setting, providing insights into its capability to scale to visual RL tasks compared to MoW's more native approach to handling visual inputs and multi-task learning.

TDMPC-2 achieves 25.3%success rate in the same environment as MoW. The results shown in Figure 15 demonstrate that while TD-MPC2 excels in continuous control tasks, MoW offers a more scalable and parameter-efficient approach for multi-task reinforcement learning that can effectively handle both continuous and discrete action spaces, especially when high-dimensional visual inputs are involved.

**TD-MPC2 and its Challenge with Discrete Actions:** The core of TD-MPC2's methodology relies on Model Predictive Path Integral (MPPI) control, which involves optimizing over continuous action sequences to maximize expected returns. The key advantage of this approach is its ability to perform local trajectory optimization in a learned world model, where actions are treated as continuous values. The model predicts continuous action trajectories over a short planning horizon and uses these predictions to guide policy execution. The trajectory optimization process inherently assumes continuous actions, as it involves sampling action sequences from a multivariate Gaussian distribution and updating their mean and variance iteratively. The Gaussian distribution allows the model to effectively explore action space by adjusting the mean and variance of actions, and this is inherently designed for continuous action dimensions. In discrete action spaces, however, unlike the continuous action spaces where actions are represented as continuous variables that can be adjusted smoothly, discrete actions are limited to a finite set of predefined choices. This discrete nature doesn't align well with the sampling mechanism in MPC, which relies on continuous action adjustments. Efficient Monte Carlo Tree Search (MCTS) or other searching methods used in the discrete action settings would require specific adjustments for handling discrete action sets, and standard MPC planning would not scale properly without a discrete action search space method.

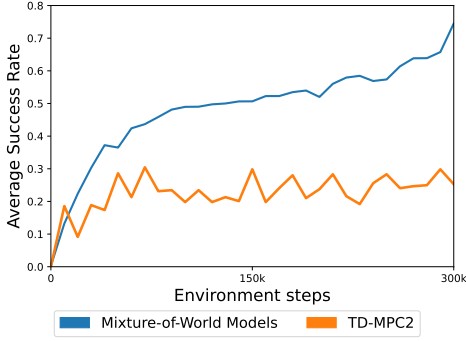

Figure 15: Comparison of MoW and TD-MPC2 on the Meta-World benchmark.

## A.12 PSEUDOCODE

For a more comprehensive understanding of training the mixture world model, we provide the pseudocode implementation in Algorithm 1.

---

**Algorithm 1** Mixture World Model-based Reinforcement Learning

---

**Input:** Task set $\mathcal{T} = \{T_1, T_2, \cdots, T_K\}$,
replay buffer $\mathcal{B} = \{B_1, B_2, \cdots, B_K\}$,
warm-up steps $T_W$
total number of environment steps $T_{total}$.

**Initialize:** Trained mixture world model parameters $\phi$,
actor–critic parameters $\theta$ and $\psi$,
actor–critic policy $\pi_\theta$,
replay buffer $\mathcal{B} = \emptyset$,
number of environment steps $t = 0$
number of warmup steps $t_w = 0$

*// Warmup stage*
**Repeat:**
    *// 1. Collect data from environment*
    $B_t^k = \{o_k^t, r_k^t, a_k^t, c_k^t\}$
    $t+ = 1$
**Until** Each $B_k$ has sufficient data for warmup training.

**Repeat:**
    *// 2. Offline training of the mixture world model*
    $D_k = $ sample-trajectory$(B_k)$
    $\phi = $ update-world-model$(D_k)$
    $t_w+ = 1$
**Until** $t_w = T_W$ .

*// Training stage*
**Repeat:**
    *// 3. Collect data from environment*
    $B_t^k = \{o_k^t, r_k^t, a_k^t, c_k^t\}$

    *// 4. Update mixture world model*
    $D_k = $ sample-trajectory$(B_k)$
    $\phi = $ update-world-model$(D_k)$

    *// 5. Sample observation for imagination*
    $O_k^1 = $ sample-obs$(B_k)$
    $\tau_k = $ world-model-imagine$(O_k^1, \pi_\theta)$

    *// 6. Learning in the imagination*
    $\theta, \psi = $ update-policy$(\tau_k, \theta, \psi)$
    $t+ = 1$
**Until** $t = T_{total}$.

---

