# OpenReview forum: "Mixture-of-World Models: Scaling Multi-Task Reinforcement Learning with Modular Latent Dynamics"
_ICLR.cc/2026/Conference — ICLR 2026 Poster_

### Official Review · Reviewer_uU94 · 2025-10-28

**Soundness:** 1
**Presentation:** 3
**Contribution:** 3
**Rating:** 4
**Confidence:** 4

**Summary:**

The Mixture-of-World Models (MoW) is a new, scalable, and parameter-efficient architecture designed for multi-task reinforcement learning (MTRL), specifically targeting visual environments where tasks have highly different visual observations and dynamics. The core challenge is that traditional model-based RL (MBRL) world models struggle to generalize across diverse tasks, leading to poor performance. MoW overcomes this by integrating three key modular components: a) Modular Variational Autoencoders (VAEs) for task-adaptive visual data compression, b) a hybrid Transformer-based dynamics model that shares a core backbone while utilizing task-conditioned specialized experts, and c) a gradient-based task clustering mechanism to efficiently allocate parameters. A single MoW agent demonstrated strong results, achieving a competitive human-normalized score of 110.4% on the Atari 100k benchmark while requiring 50% fewer parameters than the previous state-of-the-art ensemble method (STORM). It also established a new state-of-the-art success rate of 74.5% on the Meta-World benchmark, proving MoW's effectiveness as a foundation for building scalable, generalist world models.

**Strengths:**

- The paper is well-written and structured.
- The work is well-motivated: The authors clearly state the general problem of multi-task RL (MTRL) with visual inputs. They effectively motivate their proposed solution of using model-based RL (MBRL) via a world model, specifically addressing the technical obstacle of learning diverse task dynamics with a single model through the use of a Mixture of Experts (MoE).
- The learning algorithm integrates several interesting components, such as the annealing of the temperature coefficient in the softmax function, which is used to prevent distribution collapse and limit parameter sharing.
- The results on Meta-World are strong, with MoW's performance using visual inputs clearly outperforming its results using state vectors. However, I would have expected the baselines to be benchmarked with visual inputs as well, in order to demonstrate their failure to handle high-dimensional visual inputs.

**Weaknesses:**

- My main criticism is the poor empirical evaluation, which makes the submission incomplete.
- Although the two benchmarks, Atari 100k and Meta-World, are diverse, the baselines provided are very sparse.
- For instance, the learning curves for single-task STORM [1] are missing in Figure 3, where only the MoW results are shown.
- Since TD-MPC2 [2] was highlighted in the related work section, I would have expected to see it benchmarked against the proposed algorithm on both benchmarks.
- Since the baseline results on Meta-World were taken from MOORE [3], there are two issues to address.
- One issue is the total number of environment steps for the baselines. The appendix of MOORE stated 100M (2M per task), which contradicts the 20M cited here. I believe the 20M is a typo in the MOORE paper, as recently highlighted by other work [4]. Although the author is not to blame for this, the correct value should be stated.
- A more crucial issue is the mismatch between the proposed approach's hyperparameters and the MOORE baseline results (e.g., MOORE used a planning horizon of 150 instead of 500). I advise either rerunning the baselines (which I recommend) or rerunning MoW with the exact hyperparameters stated in the MOORE Appendix.
- While MoW is clearly more parameter-efficient than single-task STORM, the model size (number of parameters) for MoW and the baselines on Meta-World was not stated. I suspect that MoW uses a very large model relative to MOORE and the other baselines.
- Since the proposed algorithm contains many algorithmic components, I would have expected a full ablation study detailing their individual effect on performance.

[1] Zhang, Weipu, et al. "Storm: Efficient stochastic transformer based world models for reinforcement learning." Advances in Neural Information Processing Systems 36 (2023): 27147-27166.

[2] Hansen, Nicklas, Hao Su, and Xiaolong Wang. "Td-mpc2: Scalable, robust world models for continuous control." arXiv preprint arXiv:2310.16828 (2023).

[3] Hendawy, Ahmed, Jan Peters, and Carlo D'Eramo. "Multi-Task Reinforcement Learning with Mixture of Orthogonal Experts." The Twelfth International Conference on Learning Representations.

[4] McLean, Reginald, et al. "Multi-Task Reinforcement Learning Enables Parameter Scaling." arXiv preprint arXiv:2503.05126 (2025).

**Questions:**

- Please benchmark TD-MPC2 on both the Atari 100k and Meta-World benchmarks and compare its performance with MoW. I believe this will highlight the potential of the proposed method.
- The inclusion of the single-task STORM training curves is highly recommended. Could you please provide these for Figure 3?
- I believe there should be consistency in the learning settings across MOORE, the baselines, and MoW. Please consider updating the empirical results on Meta-World to account for these aspects.
- To strengthen the paper, it would be beneficial to demonstrate how MOORE and the other baselines fail to effectively learn from raw visual inputs, which would highlight the advantage of the proposed MoW architecture.
- What are the model sizes (e.g., number of parameters) for MoW and the baselines when benchmarking on Meta-World?
- An ablation study is requested to demonstrate the effect of changing the different algorithmic components on overall performance.
- Could you please provide the individual task learning curves in the appendix for the Meta-World tasks?

I acknowledge that these requests constitute a substantial body of work for the rebuttal period. However, addressing these crucial empirical and validation points is necessary, in my opinion, for the paper to be ready for publication and subsequent acceptance.

---

> ### Author Response · Authors · 2025-11-20
> **Respond to Weaknesses and Questions:**
>
> We sincerely thank the reviewer for the detailed feedback and recognition of our work. Below, we provide a detailed response to each point:
>
> **1. Benchmarking TD-MPC2**
>
> We recognize that comparing MoW with TD-MPC2 is crucial, especially given the importance of TD-MPC2 in the related work section. However, we must clarify that TD-MPC2 cannot be reproduced in the Atari environment with discrete action spaces. Since the core methodology of TD-MPC2 is based on Model Predictive Path Integral (MPPI) control, which optimizes continuous action sequences to maximize expected returns. The method works by sampling action sequences from a multivariate Gaussian distribution and iteratively updating their mean and variance. As stated by **TD-MPC2's authors**, TD-MPC2 cannot effectively handle environments with discrete action spaces without additional adjustments, such as Monte Carlo Tree Search (MCTS) [1]. We attempted to adapt it to the Atari environment using techniques like continuous action discretization, but still faced difficulties in learning effective policies. Furthermore, the official implementation of TD-MPC2 only includes **online training settings for visual single-task tasks** in the DMC environment and **offline multi-task training for state-based settings**. Therefore, we had to manually adapt its visual single-task configuration to the multi-task scenario faced by MoW. We provide the comparison results with TD-MPC2 in Appendix A11, where TDMPC-2 only achieves $25.3$\% success rate in the same environment as MoW.
>
> [1] Hansen, Nicklas, Hao Su, and Xiaolong Wang. "Td-mpc2: Scalable, robust world models for continuous control." arXiv preprint arXiv:2310.16828 (2023).
>
> **2. Single-task STORM**
>
> We provide the single-task learning curves reproduced from the complete STORM official implementation in Figure3. Since STORM only provides an official implementation in the Atari discrete action space, our adaptation of STORM to the continuous action setting in the Meta-World environment is primarily based on DreamerV3. Due to the computational resource constraints, these curves are still in training and evaluation.
>
> **3. Learning Settings**
>
> We appreciate the reviewer’s reminder and have updated the specific evaluation details in Table 1. Specifically, due to time and computational resource constraints, we initially chose to reproduce TD-MPC2 in our baselines, which is capable of handling visual environments, and conducted tests comparing it with MoW in the same environment. We are further adding MOORE experiments during the discussion phase.
>
> **4. Why Visual Tasks are more Difficult**
>
> For the same reason as in the third point, we use the results of TD-MPC2 method for illustration. Specifically, TD-MPC2 is tested on the DMC tasks with the same size MLP, where the only difference between the visual and state settings is the addition of a shallow convolutional network in the visual setting. The experimental results  (in Appendix A4) show that the training convergence speed and final performance with state inputs (an average score of 911) are significantly stronger than those with visual inputs  (an average score of 729). Therefore, we believe that making decisions based on visual inputs is more challenging than making decisions based on state inputs.
>
> **5. Model Sizes of MoW**
>
> The model size of MoW's world model is 1.33GB, which is larger than TDMPC2 (41MB) but smaller than MOORE (3.66GB). However, we would like to point out that directly comparing model sizes among MoW and other methods may not be appropriate, as the visual module (VAE) itself occupies a significant portion of the model's capacity. Additionally, baselines like MOORE are based on model-free methods like SAC, which do not require explicit modeling of the environment, and therefore have much lower sample efficiency compared to MoW. The primary reason MoW uses a larger model is that it needs to explicitly learn multi-task dynamics, which brings three advantages: higher sample efficiency, higher training ratio, and improved decision-making performance. In contrast to the implicit world modeling approach of TD-MPC2, even though TD-MPC2 uses a smaller model, our comparative experiments show that MoW outperforms TD-MPC2 in terms of performance with a limited number of interactions.
>
> **6. Ablation Studies and Metaworld Curves**
>
> In the latest version of the manuscript, we have included the ablation study for all components of MoW in Appendix A5. The results are also summarized in the following part of comment. Additionally, we have added the individual task curves for Meta-World in Appendix A10.2.
>
> We would like to sincerely thank you once again for your valuable suggestions to help improve our work. We hope we have addressed all your raised points. If there are anything unclear, we would be happy to provide further clarifications.

---

> ### Author Response · Authors · 2025-11-21
> **Ablation Studies**
>
> We provide additional ablation on all of our design choices for MoW in this part, including the clusters of VAE-predictors, number of experts, cascaded expert-shared transformers, task predictor head, balanced loss, and the gradient-based task clustering. The results are summarized in the tables below.
>
> 1. Clusters of VAE-predictors:  We ablate the number of clusters of VAE-predictors $N\_m$ from 12 to 3 and observe a clear performance degradation as the clusters count decreases. With fewer clusters, each VAE-predictor must capture the visual characteristics of a larger set of tasks, which slows down convergence and leads to lower overall performance.
> |Number of Clusters| 12(MoW)| 9|  6| 3|
> |:--------------|:------:|:------:|:------:|:------:|
> | Atari 100k score|110.4%|85.2%| 64.3%| 53.4%|
>
> 2. Number of Expert Transformers: We ablate the number of  expert Transformers $N_e$ from $12$ to $3$ and observe a clear performance degradation as the experts count decreases. With fewer experts, it becomes more difficult to capture the diverse dynamics across tasks, which reduces the quality of imagined trajectories generated by the world model and ultimately makes policy learning more challenging.
> |Number of Experts| 12(MoW)| 9|  6| 3|
> |:--------------|:------:|:------:|:------:|:------:|
> | Atari 100k score|110.4%|83.5%| 52.3%| 13.2%|
>
> 3. Expert and Shared Transformers: We ablate the shared Transformer and observe a modest performance drop, as tasks in Atari 100k share relatively little inter-task structure. In contrast, removing the expert Transformer leads to a much larger degradation, since the expert Transformer is responsible for capturing task-specific dynamics and generating accurate multi-task imagined trajectories.
> | | Expert + Shared(MoW)| No shared| No expert|
> |:--------------|:------:|:------:|:------:|
> | Atari 100k score|110.4%|98.5%| 15.7%|
>
> 4. Task Prediction: We ablate the task predictor by removing both the prediction head and its associated loss. The task predictor plays an important role in maintaining task-discriminative structure in the learned latent space, which benefits both world-model learning and downstream RL performance. Consistent with this, removing the task predictor leads to a noticeable decline in overall performance.
> | | MoW| No task prediction|
> |:--------------|:------:|:------:|
> | Atari 100k score|110.4%|76.3%|
>
> 5. Harmonious Loss: We ablate the harmonious loss by removing the harmonious weight. Without this operation, the world-model training faces the gradient conflicting problem, and the end-to-end loss is dominated by tasks with larger loss magnitudes, leading to unstable training and reduce the RL performance. This confirms that harmonizing the multi-task losses is important for stable and effective multi-task visual world-model learning.
> | | MoW| No hamonious loss|
> |:--------------|:------:|:------:|
> | Atari 100k score|110.4%|69.9%|
>
> 6. Balance Loss: To evaluate whether the expert-balance term is necessary to prevent overuse of a few experts, we remove the balance loss and observe that the training becomes sensitive to initialization. The balance loss is critical for ensuring all experts are utilized and prevents collapse into degenerate expert usage.
> | | MoW| No hamonious loss|
> |:--------------|:------:|:------:|
> | Atari 100k score|110.4%|102.0%|
>
> 7. Gradient-based Task Clustering: We use randomly partition tasks into the same number of clusters ($N_m=12$) to ablate the gradient-based warmup clustering. The results show that the gradient-based clustering is necessary for allocating shared parameters meaningfully and avoiding interference across unrelated tasks.
> | | MoW| Randomly task clustering|
> |:--------------|:------:|:------:|
> | Atari 100k score|110.4%|99.1%|
>
> We hope that the expanded explanations of our ablation design can help further address your concerns.

---

> ### Author Response · Authors · 2025-11-28
>
> To further address the reviewer’s concern, we have completed the evaluation curves of the single-task STORM on Atari100k and MetaWorld, and have added the results to the latest version of the manuscript (Appendix A10). Additionally, we conducted a 15M-step test of MOORE using the environment parameters for MoW in MetaWorld, which was built using the mushroom_rl environment class (with the official implementation of MOORE). The state success rate of MOORE was 43.1%. We sincerely appreciate you taking the time to acknowledge our rebuttal. Please kindly let us know if further details are required.

---

### Official Review · Reviewer_ujJn · 2025-10-30

**Soundness:** 3
**Presentation:** 2
**Contribution:** 2
**Rating:** 4
**Confidence:** 4

**Summary:**

This paper proposes a well-motivated architecture—Mixture-of-World Models (MoW)—for multi-task visual RL that marries task-specific VAEs with a mixture-of-expert Transformers and a shared backbone, plus task-prediction and expert-balance losses and a gradient-based task clustering warm-up. The approach targets the core pain point of heterogeneous observations/dynamics and reports strong results: a single model reaches 110.4% human-normalized score on Atari-100k, competitive with STORM’s 114.2% while using ~50% fewer parameters, and achieves 74.5% average success on Meta-World MT50 within 300k steps per task. Overall, this is a solid contribution to scalable, parameter-efficient world models for MTRL with promising empirical evidence.

**Strengths:**

**Clear modular design**. MoW uses reasonable modular architecture designs.

**Promising parameter scalability**. MoW demonstrates promising parameter scaling capability. However, it would be interesting if the author can further scale the parameters to investigate if the performance could be further exponentially improved.

**Weaknesses:**

**Natation clarity**. It would be beneficial to clarify the notations in the figures, for example, Fig. 1, which could significantly enhance the readability of this paper. It would also be beneficial if you could explain the notation a little bit when it first appears. Details see questions.

**Lack of experimental evidence**. The author didn't demonstrate the common Atari-100k performance table, including mainstream baselines.

**Insufficient baselines.** STORM is actually not a multi-task RL baseline. I personally cannot understand why the authors include more single-task baselines as comparisons.

**Questions:**

1. What does $m_{\phi,j}$ represent in Equation 2?
2. Is $W_k$ sorted? To my perspective, it is not, so you should verify the definition of $W_k$.
3. What does $N_m$ represent?
4. What does $l_k^{1:t}$ represent?

---

> ### Author Response · Authors · 2025-11-20
> **Respond to Weaknesses and Questions:**
>
> We sincerely appreciate the reviewer’s thoughtful feedback and would like to address the points raised below.
>
> **1. Natation Clarity**
>
> We thank the reviewer for pointing out the need to clarify the notations in the figures, which will indeed enhance the readability and comprehensibility of the paper. In response, we have revised Figure 1 and other relevant sections to provide clearer and more explicit explanations of the notations used. We have ensured that all symbols are introduced and explained at their first occurrence in the main text, allowing readers to better follow the discussion. These changes will be highlighted in blue in the latest version of the manuscript.
>
> **2. Experimental Evidence**
>
> We  include the comprehensive performance table for the Atari-100k benchmark, comparing MoW with mainstream transformer-based baselines in the latest version of manuscript. The evidence is shown in Appendix A10.
>
> **3. Baselines**
>
> One reason we chose STORM as our baseline is that MoW draws inspiration from STORM's modeling and training setup. Therefore, using STORM as a baseline allows us to better highlight the advantages of MoW’s components. In response to the concern about using STORM as a baseline, we have reproduced the visual multi-task version of TD-MPC2 in Appendix A.11 and conducted comparative experiments with MoW under the same environmental setup. The results show that while TD-MPC2 performs excellently in state-based multi-task settings, its performance under visual multi-task conditions with $15$M interactions (a success rate of 25.3%) is nearly 50% lower than that of MoW (a success rate of 74.5%).
>
> **4. Symbols in Questions**
>
> $m_{\phi, j}$  is the MLP used to concatenate the stochastic representation $z_k^t$ and action $a_k^t$. $W_k$ is not sorted and the position of non-zero items in $W_k$ indicates the activation of the corresponding expert. $N_m$ is the number of VAEs, distribution, reward, continuation predictors and critic networks. $l^{1:t}_{k}$ is the token concatenated by output from activated expert transformers. We also highlight these explanation in blue in the latest version of the manuscript.
>
>
> **5. Further Scale**
>
> We greatly appreciate the reviewer’s positive feedback on MoW’s modular design and promising parameter scalability. However, we must acknowledge that the scalability of the parameters is not indefinite. In multi-task settings, its upper limit will eventually approach the performance level of single-task settings [1]. This is also why we initially chose single-task STORM as a baseline, as MoW comes very close to the single-task performance at a highly competitive level.
>
> [1] Sodhani, Shagun, et al.  Multi-task reinforcement learning with context-based representations. ICML 2021.
>
> We hope these clarifications address your concerns. We would be glad to provide further details if needed.

---

> > ### Comment · Reviewer_ujJn · 2025-11-24
> > **Insufficient performance on Atari 100k.**
> >
> > Thanks for authors' response. MoW is a model-based RL approach. In this regard, authors should expand the baseline set to include more up-to-date algorithms like EfficientZero, BBF, especially on Atari-100k. MoW cannot even show a significant performance increment compared with STORM and outdated IRIS according to Table 11. Therefore, I think this paper cannot prove the efficacy and superiority of MoW.

---

> ### Author Response · Authors · 2025-11-25
> **Response to the performance:**
>
> We appreciate the reviewer’s feedback regarding the comparison between MoW and baseline models over STORM and IRIS. In response, we have added the EfficientZero (194.5% of human mean score) and BBF (224.7% of human mean score) baselines to Table 11. At the same time, we would like to clarify that EfficientZero utilizes lookahead search methods, which typically incur more computational overhead and are orthogonal to our work. Therefore, it is not generally discussed alongside algorithms that focus on world model construction [1]. In contrast, BBF is a **model-free RL** method that fundamentally relies on self-supervised prediction to enhance encoding capabilities, and does not explicitly involve the construction of a world model.
>
> Furthermore, we would like to clarify that, unlike single-task models such as STORM and IRIS, MoW is designed for a multi-task setting, where **a single agent** is tasked with **learning and solving all 26 games in the Atari-100k benchmark**. This presents a more complex challenge, as the agent must generalize to a diverse range of tasks, each with unique dynamics, visual inputs, and reward structures, while maintaining sample efficiency. Particularly in the case of Atari-100k, where the correlation between different tasks is minimal, it is challenging for a single agent to generalize across all tasks [2]. To the best of our knowledge, we are the first work to test a world model-based online reinforcement learning approach in a multi-task setting on Atari-100k. In contrast,single-task models like STORM (**each task is associated with its own individual agent and 26 world models/agents for 26 tasks**) are focused on single-task learning and do not support multi-task setups (as we shown in the ablation studies Sec 5). When comparing MoW with single-task models, the task that MoW addresses is inherently more complex, requiring the agent to optimize and balance multiple tasks simultaneously. Nevertheless, MoW (110.4% of human mean score) with a single agent still outperforms the classic baselines TWM (96% of human mean score) and IRIS (105% of human mean score) under the same sample efficiency.
>
> It is also worth noting that the challenges faced by MoW go beyond training a single agent to generalize across all 26 Atari-100k tasks. The world model must also account for **modeling the heterogeneous dynamics and varying visual inputs of the multi-task environment** to enhance sample efficiency. In fact, the multi-task setting with visual inputs that MoW faces is more challenging than the state-based multi-task settings typically used in traditional multi-task reinforcement learning methods. For example, TDMPC2 with visual inputs converges more slowly and performs worse compared to TDMPC2 with state-based inputs (729.02 vs 911.05, the results and analysis are also shown in Appendix A11). At the same time, **MoW achieved superior performance (a success rate of 74.5%) compared to classic multi-task method TDMPC2 (a success rate of 25.3%) on MetaWorld's multi-task evaluation with higher sample efficiency (300k steps per task)**. Given the inherent difficulty of multi-task learning, even achieving performance comparable to single-task models underscores MoW's potential. We believe that MoW’s multi-task setting provides valuable insights into parameter efficiency and scalability to a large number of tasks, representing an important step forward in multi-task reinforcement learning.
>
> We hope this response adequately addresses the reviewer's concerns and further clarifies the challenges and contributions of MoW. We sincerely appreciate the opportunity to discuss these points and are grateful for your time and consideration in reviewing our additional clarifications.
>
> [1] Burchi M, et al. Learning transformer-based world models with contrastive predictive coding. ICLR, 2025.
>
> [2] Bellemare M G, et al. The arcade learning environment: An evaluation platform for general agents[J]. Journal of artificial intelligence research, 2013, 47: 253-279.

---

> ### Author Response · Authors · 2025-11-28
> **Further analysis of the multi-task setting:**
>
> As the discussion period is nearing its end with less than a week remaining, we would like to take this opportunity to further demonstrate the value and performance of MoW, while also addressing your concerns. Therefore, we would like to further introduce the challenges of the visual multi-task setting. As we demonstrated in the ablation studies Sec 5, directly extending a **single-task method** to a **multi-task setting** results in a significant drop in the performance. For instance, a simple extension of the single-task world model, STORM, to multi-task settings in Atari only achieves 12.3% of the human normalization score (where single-task STORM achieves 114.2%). Additionally, we have included supplementary experiments based on the classical multi-task algorithm, TDMPC2 (in Appendix A4). In the visual and state tasks of TDMPC2 on the DMC suite, we observed that the convergence of the visual tasks was not only significantly slower than that of the state tasks, but the final performance was also much harder to improve (an average score of 911 for state-input and an average score of 729 for visual-input). At the same time, the superiority of MoW also lies in the fact that, as emphasized in the comment before, MoW achieves 26 tasks with** a single agent**, approaching the single-task baseline while outperforming the classical baseline that does not use MCTS. We hope the above clarification further underscores the value of MoW, meanwhile, we are eager to address any remaining issues to improve our work.
>
> Thank you for your time and effort in reviewing our paper.

---

### Official Review · Reviewer_hP6A · 2025-10-31

**Soundness:** 3
**Presentation:** 2
**Contribution:** 2
**Rating:** 2
**Confidence:** 4

**Summary:**

This paper proposes Mixture-of-World Models (MoW) for multi-task reinforcement learning. The architecture employs task-cluster-specific VAEs combined with a hybrid transformer containing both shared and expert modules. Task grouping is performed through gradient-based clustering, and the training incorporates an expert balance loss and adaptive loss weighting to stabilize learning. Experiments on Atari100k and Meta-World benchmarks show that MoW achieves comparable or improved performance with a more compact model design.

**Strengths:**

1. To the best of my knowledge, modular architectures for world models are still underexplored, making this paper’s direction novel and valuable.
2. The design choices—such as task-level routing, the combination of fixed task clusters for VAEs with dynamic expert routing for transformers—are conceptually interesting and insightful.
3. The qualitative comparison in Figure 2 shows that MoW produces notably higher-quality imagined rollouts than the multi-task STORM baseline.

**Weaknesses:**

1. **The ablation study is severely incomplete.** The method introduces many components—such as cluster-specific VAEs, cascaded expert-shared transformers, an auxiliary task predictor head, balanced loss, and gradient-based task clustering—but none of these components are ablated to clarify their individual contributions to performance. Without such analysis, it is hard to assess which design elements are truly essential.
2. The overall architecture is highly complex, introducing a large number of new hyperparameters, which may make reproducibility and tuning difficult.

**Questions:**

1. While there are few studies on online multi-task reinforcement learning on Atari, there exist some offline multi-task works [1, 2]. Could MoW be extended to offline settings, and would its modular design still provide efficiency or performance advantages there? (I understand this would be a substantial effort and not necessarily feasible during the review period.)
2. In Line 372, how exactly is the gradient vector per task computed? Is it obtained by averaging gradients over all mini-batches associated with each task?

[1] Multi-Game Decision Transformers

[2] Scaling Offline Model-Based RL via Jointly-Optimized World-Action Model Pretraining

---

> ### Author Response · Authors · 2025-11-20
> **Respond to Weaknesses and Questions:**
>
> We appreciate your valuable feedback and the recognition of the value of our algorithm. We address each concern below.
>
> **1. Ablation Study**
>
> We greatly appreciate your feedback regarding the incomplete ablation study. In the latest version of the manuscript, we have expanded the ablation analysis in the next part of comment (Appendix A.5 of the manuscript), where we ablate and analysis the impact of each component of MoW on overall performance, thereby providing a more comprehensive and refined ablation study. This revision better showcases the distinct characteristics and contributions of MoW.
>
> **2. Hyperparameters**
>
> We significantly reduce the number of hyperparameters through a highly modular design. Beyond the common hyperparameters used in single-task world models, the main hyperparameters we consider are: the number of VAE-predictors-critics $N_m$,  the number of expert transformers $N_e$, and the number of activated experts $N_k$. To minimize the number of hyperparameters as much as possible, we use gradient clustering to assign a fixed set of VAE, predictors and critic networks to each task (as highlighted in lines 175-177 of the latest version of the paper). These three parameters can be selected based on visual similarity, task similarity, and task complexity, and are common hyperparameters in MoE and multi-task learning.
>
> **3. Offline Settings**
>
> We appreciate the reviewer’s suggestion to extend MoW to offline settings. We believe that one of the main advantages of offline reinforcement learning lies in its ability to leverage large amounts of expert data for pre-training, enabling world models to quickly initialize on reward-sparse tasks with zero success rates. Recent work has demonstrated that efficient offline policy training using single-task  world models [1]. MoW is an effective extension of these single-task world models to multi-task settings, designed primarily for multi-task modeling within world models. It does not have the limitations or special requirements for policy training, and thus can be effectively generalized from single-task world models. Therefore, extending MoW to offline settings is a feasible approach. However, despite the efficiency gains world models offer in terms of sample efficiency, offline settings still require computational resources and training time orders of magnitude greater than online training, making it challenging to fully explore during the review phase.
>
> [1] Danijar Hafner, et al. Training agents inside of scalable world models. arXiv 2025.
>
> **4. Gradient Vector Computing**
>
> We use torch.autograd to collect the gradient vector for each trainable parameter of the world model for every task's batch of data, and flatten these gradients into a gradient vector. This approach is inspired by multi-task learning [2]. During the training process, we only need to collect the gradient data from the final step of the warmup phase for clustering, as the gradients converge very quickly during the warmup phase. We show the visualization of this process in Appendix A3.
>
> [2] Hu, Shengchao, et al. Harmodt: Harmony multi-task decision transformer for offline reinforcement learning. arXiv 2024.
>
> We hope these clarifications address the reviewer’s concerns. Please let us know if there are any remaining questions or aspects that could benefit from further clarification.

---

> ### Author Response · Authors · 2025-11-21
> **Ablation Studies**
>
> We provide additional ablation on all of our design choices for MoW in this part, including the clusters of VAE-predictors, number of experts, cascaded expert-shared transformers, task predictor head, balanced loss, and the gradient-based task clustering. The results are summarized in the tables below.
>
> 1. Clusters of VAE-predictors:  We ablate the number of clusters of VAE-predictors $N\_m$ from 12 to 3 and observe a clear performance degradation as the clusters count decreases. With fewer clusters, each VAE-predictor must capture the visual characteristics of a larger set of tasks, which slows down convergence and leads to lower overall performance.
> |Number of Clusters| 12 (MoW)| 9|  6| 3|
> |:--------------|:------:|:------:|:------:|:------:|
> | Atari 100k score|110.4%|85.2%| 64.3%| 53.4%|
>
> 2. Number of Expert Transformers: We ablate the number of  expert Transformers $N_e$ from $12$ to $3$ and observe a clear performance degradation as the experts count decreases. With fewer experts, it becomes more difficult to capture the diverse dynamics across tasks, which reduces the quality of imagined trajectories generated by the world model and ultimately makes policy learning more challenging.
> |Number of Experts| 12 (MoW)| 9|  6| 3|
> |:--------------|:------:|:------:|:------:|:------:|
> | Atari 100k score|110.4%|83.5%| 52.3%| 13.2%|
>
> 3. Expert and Shared Transformers: We ablate the shared Transformer and observe a modest performance drop, as tasks in Atari 100k share relatively little inter-task structure. In contrast, removing the expert Transformer leads to a much larger degradation, since the expert Transformer is responsible for capturing task-specific dynamics and generating accurate multi-task imagined trajectories.
> | | Expert + Shared (MoW)| No shared| No expert|
> |:--------------|:------:|:------:|:------:|
> | Atari 100k score|110.4%|98.5%| 15.7%|
>
> 4. Task Prediction: We ablate the task predictor by removing both the prediction head and its associated loss. The task predictor plays an important role in maintaining task-discriminative structure in the learned latent space, which benefits both world-model learning and downstream RL performance. Consistent with this, removing the task predictor leads to a noticeable decline in overall performance.
> | | MoW| No task prediction|
> |:--------------|:------:|:------:|
> | Atari 100k score|110.4%|76.3%|
>
> 5. Harmonious Loss: We ablate the harmonious loss by removing the harmonious weight. Without this operation, the world-model training faces the gradient conflicting problem, and the end-to-end loss is dominated by tasks with larger loss magnitudes, leading to unstable training and reduce the RL performance. This confirms that harmonizing the multi-task losses is important for stable and effective multi-task visual world-model learning.
> | | MoW| No hamonious loss|
> |:--------------|:------:|:------:|
> | Atari 100k score|110.4%|69.9%|
>
> 6. Balance Loss: To evaluate whether the expert-balance term is necessary to prevent overuse of a few experts, we remove the balance loss and observe that the training becomes sensitive to initialization. The balance loss is critical for ensuring all experts are utilized and prevents collapse into degenerate expert usage.
> | | MoW| No hamonious loss|
> |:--------------|:------:|:------:|
> | Atari 100k score|110.4%|102.0%|
>
> 7. Gradient-based Task Clustering: We use randomly partition tasks into the same number of clusters ($N_m=12$) to ablate the gradient-based warmup clustering. The results show that the gradient-based clustering is necessary for allocating shared parameters meaningfully and avoiding interference across unrelated tasks.
> | | MoW| Randomly task clustering|
> |:--------------|:------:|:------:|
> | Atari 100k score|110.4%|99.1%|
>
> We hope that our detailed ablation settings can further address your concerns.

---

> ### Author Response · Authors · 2025-11-28
>
> We have further included the curves for all ablation experiments mentioned in the rebuttal in Appendix A5. We appreciate the time and effort that the reviewers have put into evaluating our work. Given the upcoming deadlines, we are eager to address any remaining concerns and ensure the timely completion of this process. If there are anything unclear, we would be happy to provide further clarifications. Thank you once again for your attention to our submission.

---

### Official Review · Reviewer_XJhM · 2025-11-01

**Soundness:** 3
**Presentation:** 3
**Contribution:** 3
**Rating:** 6
**Confidence:** 2

**Summary:**

This paper addresses the challenge of sample efficiency in Multi-Task Reinforcement Learning (MTRL). The authors propose Mixture-of-World (MoW) Models, a modular architecture integrating:
- Modular VAEs specifically clustered for task-adaptive visual compression.
- A hybrid dynamics model combining task-conditioned MoE (Mixture of Experts) Transformers with a shared Transformer backbone.
- Gradient-based task clustering to efficiently allocate parameters during a warmup phase.

**Strengths:**

1. Tailored MoE specifically for World Model Dynamics. Standard MoE in Large Language Models often uses token-level routing. The authors rightly identify that for dynamics modeling, token-level routing can lead to "fragmented learning," where sequential temporal dependencies are broken if consecutive tokens route to different experts.
2. End-to-end MTRL often suffers from gradient conflicts and tasks dominating the loss landscape. MoW integrates distinct mechanisms to stabilize this.
3. Parameter Efficiency via Gradient Clustering. Instead of naively assigning a VAE per task (expensive) or one VAE for all (insufficient for heterogeneous visuals), they use a warmup phase to cluster tasks based on gradient similarity. This allows intelligent parameter sharing for the perceptual modules, balancing capacity with efficiency

**Weaknesses:**

1. Dependency on Warmup Quality. The gradient-based clustering relies heavily on a "warmup stage" where a single VAE/predictor set is trained. If the initial warmup yields noisy gradients (common in early RL from pixels), the resulting clusters might be suboptimal and fixed for the rest of training. The paper does not deeply analyze the sensitivity of final performance to the duration or stability of this warmup phase.
2. Architectural Complexity and Tuning. The system is highly complex, involving multiple specialized losses (reconstruction, reward, continuation, task prediction, dynamics KL, representation KL, harmonious weighting, expert balance). While effective, this increases the hyperparameter surface significantly. Reproducibility outside specifically tuned benchmarks might be challenging without robust auto-tuning mechanisms for these loss components.
3. Routing Rigidity. While task-level routing prevents temporal fragmentation, it might be too rigid for tasks that share partial sub-dynamics. Task-level routing forces it to choose one set of experts for the entire episode.

**Questions:**

1. Given the high variance of gradients early in standard RL training (especially from pixels), how stable are the resulting task clusters across different random seeds? Did you observe cases where a poor warmup led to irrecoverable sub-optimal clusterings?
2. How sensitive is MoW to the fixed hyperparameters? Were these tuned specifically for Atari and Meta-World, or do you expect them to generalize to new domains?
3. Does task-level routing limit zero-shot generalization to new tasks? If a new task is a composition of two existing tasks, your current architecture forces it to choose just one set of experts, whereas token-level might allow it to interpolate.

---

> ### Author Response · Authors · 2025-11-20
> **Respond to Weaknesses and Questions:**
>
> We sincerely thank you for the detailed and constructive feedback. We address each concern below.
>
> **1. Dependency on Warmup Quality**
>
> We would like to clarify that, as we shown in the pseudocode (Appendix. A12), the training during the warmup phase is exclusively focused on the world model itself, without involving policy training. This design avoids the issue of noisy gradients typically encountered during the early stages of RL. Additionally, since the warmup stage is conducted on the fixed dataset collected during the warmup period, the MoW model, which is capable of multi-task modeling, converges efficiently to a stable state within just a few thousand steps (as we shown in Appendix A3). Since sufficiently stable gradient vectors are collected during the warmup stage, the gradient-based clustering remains sufficiently stable, which is demonstrated in ablation studies Appendix A5, using random clustering fails to match the performance of the task-specific gradient clustering method..
>
> **2. Hyperparameter Sensity**
>
> The weight configuration for the reconstruction, reward, continuation, dynamics, and representation losses in MoW is consistent with that of most world model algorithms, as single-task world models aim to use the same hyperparameters across different tasks [1,2]. Therefore, these losses  simply carry over the settings from single-task world models.and are not specifically designed for MoW. The task prediction loss, on the other hand, is an additional loss introduced to enhance the representational capacity of the hidden states. It converges to a stable value early in training, functioning similarly to a small regularization term and is not sensitive to the weights. The harmonious loss is designed for multi-task settings, with all harmonic weights initialized to 1. As we proved in Appendix A7, these weights adaptively converge to a unified scale during training, helping to mitigate gradient conflicts and prevent large task losses from disproportionately influencing the overall loss. The harmonious weights are automatically trained and do not require manual adjustment. The balance loss is intended to encourage balanced routing by selecting experts, preventing imbalances in expert load. As we shown in the ablation study in Appendix A5, MoW is not sensitive to the balance loss. Therefore, we would like to emphasize that the hyperparameters of MoW are not specifically designed for Atari or Meta-World, but are intended to be transferable to new domains.
>
> [1] Danijar Hafner, et al. Mastering diverse control tasks through world models. Nature 2025.
>
> [2] Zhang, Weipu, et al. Storm: Efficient stochastic transformer based world models for reinforcement learning. NeurIPS 2023.
>
> **3. Routing Rigidity**
>
> As we discussed in Sec 3.2, the primary reason we do not use token-level routing is due to the temporal fragmentation. For world model-based RL, the ability to model continuous time significantly impacts the quality of imagined trajectories, which directly affects the policy's performance. Temporal fragmentation caused by token-level routing would prevent the attention mechanism from effectively capturing dynamic relationships in a multi-task setting, leading to a collapse in the quality of imagination trajectories. Regarding the issue of routing rigidity, we would like to clarify that our decision-making is handled by a single agent (although it is equipped with multiple Critic networks for different tasks). The MoW agent itself learns decision-making knowledge across multiple tasks, allowing it to handle situations where a new task is composed of two existing tasks. At the same time, the task-level routing determined by task embeddings is also trainable. This design strikes a balance between flexibility and stability, allowing the model to efficiently learn across a variety of tasks while preserving the integrity of task-specific dynamics.
>
> We appreciate the reviewer’s thoughtful feedback and hope that our rebuttal has addressed your concerns. Please kindly let us know if further details are required.

---

> ### Author Response · Authors · 2025-11-28
>
> To further address the concern about the warmup quality, we have further added the curves for the ablation study about the randomly clustering referenced in the rebuttal to Appendix A5 (110.4% of gradient-based clustering and 99.1% of randomly clustering). We sincerely appreciate the reviewers' time and effort in evaluating our work. We are eager to address any remaining questions or concerns. Should any aspects require further clarification, we would be happy to provide additional details. Thank you again for your attention to our submission.

---

### Meta-Review · Area_Chair_Mdo9 · 2026-01-06

**Summary:**

The paper introduces a mixture-of-experts architecture for multi-task model based RL with visual inputs. In essence, by learning to assign different experts to different tasks, better parameter efficiency is obtained.

The contribution is mostly technical -  how to design and train a transformer model with multiple experts to make the best use of the data, and the ideas include keeping the expert selection consistent throughout each task, and tricks for balancing the different losses during training. While the technical ideas are rather simple, their successful combination, as reflected in the results, is a worthy finding to report.

One thing that bothered me is the focus on a fixed number of tasks, without generalization to new tasks (although since there are more tasks than experts this is not trivial). But this setting has been explored before in this very specific area of RL.

**Reviewer Concerns:**

The reviewers acknowledged the novelty of MoE approach to multi-task model based RL. Most concerns were technical: parameter sensitivity, comparison to baselines, and ablation studies. The authors supplied those in the rebuttal. There were concerns that SoTA model based methods for Atari outperform the current approach, which the authors explained is due to the use of online search (MCTS). I agree that the current investigation is different enough to make it interesting.

**Reviewer Scores:**

6,2,4,4

Most concerns were addressed in the rebuttal, and I believe the scores would have gone up. The reviewers did not respond yet to the rebuttal.

I base my decision on the reviews, the rebuttal, and on my own reading and reviewing of the paper.

---

### Decision · Program_Chairs · 2026-01-26

Accept (Poster)